# V-ABS: Action-Observer Driven Beam Search for Dynamic Visual Reasoning

**Zhiwei Ning** [1 2 3]  **Xuanang Gao** [1 2]  **Jiaxi Cao** [1 2]  **Gengming Zhang** [1 2]  **Shengnan Ma** [3]  **Wenwen Tong** [3]
**Hanming Deng** [3]  **Jie Yang** [1 2 4]  **Wei Liu** [1 2 4 *]

## Abstract

Multimodal large language models (MLLMs) have achieved remarkable success in general perception, yet complex multi-step visual reasoning remains a persistent challenge. Although recent agentic approaches incorporate tool use, they often neglect critical execution feedback. Consequently, they suffer from the imagination-action-observer (IAO) bias, a misalignment between prior imagination and observer feedback that undermines reasoning stability and optimality. To bridge this gap, we introduce V-ABS, an action-observer driven beam search framework that enables deliberate reasoning through thinker-actor-observer iterations. We also propose an entropy-based adaptive weighting algorithm to mitigate the IAO bias by dynamically balancing the confidence scores between the policy priors and the observational feedback. Moreover, we construct a large-scale supervised fine-tuning (SFT) dataset comprising over 80k samples to guide the model to assign higher prior confidence to correct action paths. Extensive experiments across eight diverse benchmarks show that V-ABS achieves state-of-the-art performance, delivering an average improvement of 19.7% on the Qwen3-VL-8B baseline and consistent gains across both open-source and proprietary models. Code is available at https://github.com/pami-zwning/V-ABS.

## 1. Introduction

Large language models (LLMs) have showcased remarkable capabilities in textual understanding and complex reason-

[1]School of Automation and Intelligent Sensing, Shanghai Jiao Tong University [2]Institute of Image Processing and Pattern Recognition, Shanghai Jiao Tong University [3]SenseTime Research [4]Institute of Medical Robotics, Shanghai Jiao Tong University. Correspondence to: Wei Liu <weiliucv@sjtu.edu.cn>.

*Proceedings of the 43rd International Conference on Machine Learning*, Seoul, South Korea. PMLR 306, 2026. Copyright 2026 by the author(s).

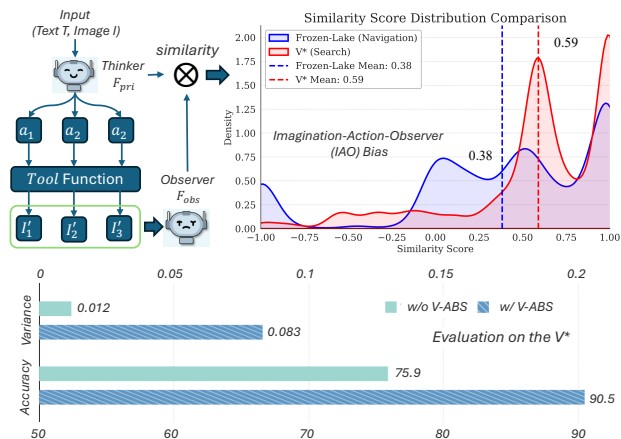

*Figure 1.* Analysis of the IAO bias. The top panel reveals a significant discrepancy between the model's prior scores $\mathcal{F}_{pri}$ and visual utility outcomes $\mathcal{F}_{obs}$. The bottom panel demonstrates that V-ABS exhibits significant variance in confidence scores across different actions, which yields a substantial accuracy gain on the V* benchmark.

ing (Touvron et al., 2023; Achiam et al., 2023). Building upon this foundation, recent multimodal large language models (MLLMs) such as LLaVA-OneVision (Li et al., 2024a), Qwen2-VL (Wang et al., 2024), and Gemini (Team et al., 2024) demonstrate exceptional proficiency in general visual perception tasks, ranging from image captioning to visual question answering (Liu et al., 2024; Bai et al., 2023b). However, a significant disparity remains between basic visual perception and complex visual reasoning. Consequently, current MLLMs often falter when confronted with multi-step reasoning tasks that necessitate dynamic interaction with visual environments, such as visual search and navigation.

Inspired by the System 2 thinking paradigm in human cognition (Kahneman, 2011), recent research attempts to emulate deliberative processes in MLLMs through test-time scaling techniques (Snell et al., 2024; Zeng et al., 2025). Although strategies like chain-of-thought prompting (Wei et al., 2022) and tree search (Yao et al., 2023; Feng et al., 2023) have been adapted for multi-step reasoning, they are often constrained by rigid interaction paradigms that operate primarily on formalized representations (Wang et al., 2025c; Zhang et al., 2023). While agentic approaches such as ZoomEye (Shen

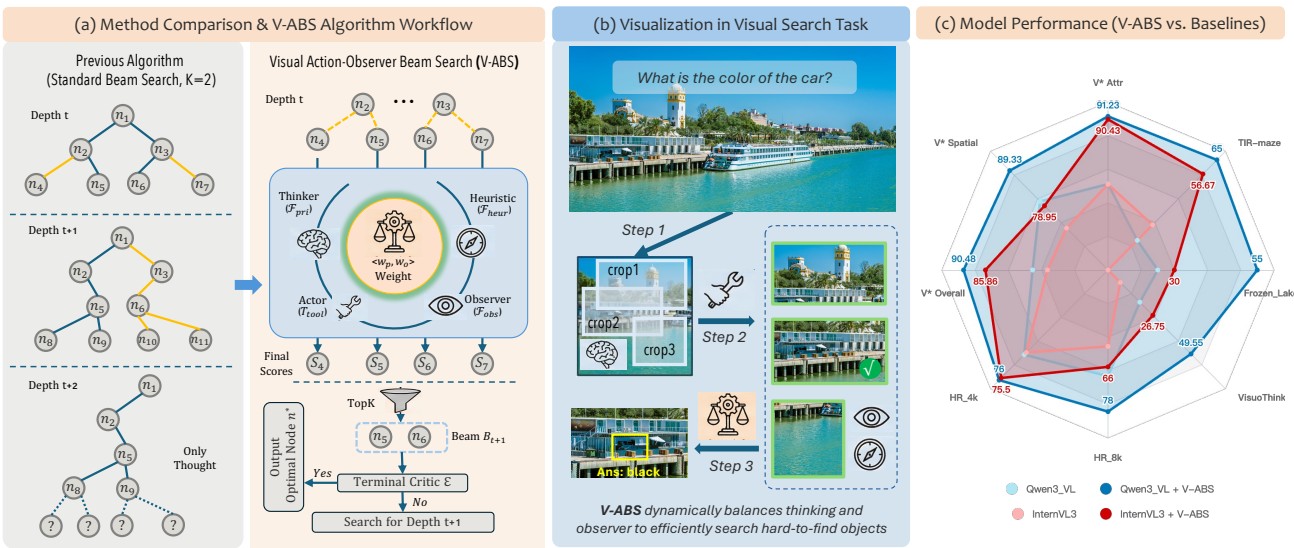

*Figure 2.* Overview of the V-ABS framework. (a) Action-observer driven algorithm: at each reasoning step, the thinker generates prior scores $\mathcal{F}_{pri}$ for each candidate action, the actor executes these actions via tool functions $\mathcal{T}_{tool}$ to update the visual state, and the observer evaluates the resulting states to obtain feedback scores $\mathcal{F}_{obs}$, assigned by a heuristic score $\mathcal{F}_{heur}$. The adaptive weighting mechanism dynamically balances these scores to ensure the optimal trajectory. (b) Visualization: an example of multi-step visual search where V-ABS actively updates the visual context through cropping operations. (c) Quantitative performance: V-ABS significantly outperforms the baseline models across diverse tasks.

et al., 2025), DeepEyes (Hong et al., 2025), and Pixel Reasoner (Wang et al., 2025a) facilitate active visual exploration via cropping or zooming, they typically select actions based on initial priors, neglecting critical post-execution feedback. Our investigation reveals that a distinct deviation exists between the preliminary confidence and the actual outcomes. As illustrated in Figure 1, we quantify the distribution of the scaled cosine similarity between the prior confidence $\mathcal{F}_{pri}$ of each action and the corresponding observed confidence $\mathcal{F}_{obs}$ after the manipulation. The result reveals a low average value of 0.59 in visual search and 0.38 in navigation. We term this phenomenon the imagination-action-observer (IAO) bias. This phenomenon indicates that action routes based only on prior thinking are unreliable, and determining the next step inevitably yields suboptimal trajectories.

To mitigate the IAO bias, we introduce V-ABS, an action-observer driven beam search framework for visual reasoning. As illustrated in Figure 2(a), V-ABS transforms thought-based beam search (Wiseman & Rush, 2016) into a closed-loop system. The framework proceeds as follows: the thinker samples and scores potential actions from candidate sets, the actor executes these actions via external tools to generate updated visual states, and the observer evaluates the utility of the resulting states, finalized by heuristic adjustments. Simultaneously, an adaptive weighting algorithm balances the prior scores $\mathcal{F}_{pri}$ and observer feedback $\mathcal{F}_{obs}$ to address the IAO bias. This mechanism transforms the model from a passive perceiver into an active agent, dynamically guiding the search toward the most promising trajectory,

as illustrated in Figure 2(b). Furthermore, we construct a large-scale dataset comprising over 80k samples across diverse domains for supervised fine-tuning (SFT). This dataset is specifically designed to encourage the model to assign higher probabilities to correct actions in the thinking stage and thereby enhance the overall inference accuracy.

Benefiting from this technique, V-ABS exhibits significant variance across candidate actions, thereby increasing the likelihood of choosing the correct path and achieving higher accuracy, as shown on the bottom of Figure 1. We also evaluate V-ABS across diverse visual reasoning datasets, as illustrated in Figure 2(c). Our method demonstrates exceptional performance and generalization, delivering average improvements of 19.7% on Qwen3-VL-8B and 13.9% on InternVL-3-8B. Crucially, V-ABS consistently outperforms the state-of-the-art methods (Zhang et al., 2025; Wang et al., 2025a) that rely on both SFT and reinforcement learning (RL) paradigms. Our contributions are summarized as:

- We introduce V-ABS, a novel framework that integrates planning, execution, and verification into a closed-loop beam search process. This architecture enables active visual manipulation and evaluation, which provides reasoning in physical reality to mitigate open-loop uncertain propagation.

- We identify the imagination-action-observer bias and introduce an adaptive weighting algorithm to balance the confidence of prior thinking and the visual observer. Moreover, we also construct a large-scale SFT dataset

to encourage the model to give higher prior scores to the correct actions in each step.

- Extensive experiments across diverse benchmarks demonstrate that V-ABS achieves state-of-the-art performance and robust generalization, delivering significant gains on both open-source and proprietary models.

## 2. Related Work

**Advancements in Multimodal Large Language Models** The landscape of multimodal large language models (MLLMs) has evolved from early alignment strategies to high-resolution native architectures. Foundational works like BLIP-2 (Li et al., 2023) and InstructBLIP (Dai et al., 2023) utilize Q-Formers to bridge frozen visual encoders with LLMs. Following LLaVA (Liu et al., 2023), the field shifted towards visual instruction tuning. To conquer resolution bottlenecks, architectures such as Qwen-VL (Bai et al., 2023a), Monkey (Li et al., 2024b), and Mini-Gemini (Li et al., 2025b) introduce high-resolution cropping and dual-encoder mechanisms. More recently, Cambrian-1 (Tong et al., 2024) and InternVL2.5 (Chen et al., 2024) have pushed the boundaries with massive-scale vision backbones and dynamic resolution (AnyRes) support. Despite these perceptual strides, most MLLMs remain bound to a System 1 paradigm, processing visual inputs as immutable tensors in a single forward pass.

**Search Algorithm for Test-Time Scaling** Recently, scaling compute at inference time has become a critical technique for visual reasoning (Wang et al., 2022). Advances like OpenAI-o1 (Jaech et al., 2024) and DeepSeek-R1 (Guo et al., 2025) demonstrate the power of prolonged thought chains. In the text domain, search algorithms have evolved from tree of thoughts (ToT) (Yao et al., 2023) to graph of thoughts (GoT) (Besta et al., 2024) and reasoning via planning (RAP) (Hao et al., 2023), using Monte Carlo tree search (MCTS) (Feng et al., 2023) or guided beam search. Methods like Multimodal-CoT (Zhang et al., 2023), DDCoT (Zheng et al., 2023), and VisuoThink (Wang et al., 2025c) attempt to extend these strategies in the multimodality area. However, existing training-free methods predominantly operate within the textual latent space or rely on passive attention cropping (Xiang et al., 2024), failing to physically update the visual information. V-ABS addresses this by incorporating a dynamic action-observer loop, extending test-time scaling into the physical visual action space.

**Visual Reasoning with Tools and Agents** To handle complex visual tasks, researchers have integrated MLLMs with external tools (Schick et al., 2023). General-purpose agents like Chameleon (Lu et al., 2023), HuggingGPT (Shen et al., 2023), and MM-REACT (Yang et al., 2023) orchestrate APIs for composite tasks. For explicit reasoning, ViperGPT (Surís et al., 2023) generates code to query visual content, while Visual Sketchpad (Hu et al., 2024) employs drawing tools. Specialized methods like ZoomEye (Shen et al., 2025) and V* (Wu & Xie, 2024) utilize zooming mechanisms for fine-grained details. Although promising, these approaches often suffer from linear execution flows or rigid heuristics, lacking a mechanism to verify if an action (e.g., $<\texttt{zoom}>$) actually yielded useful information. V-ABS improves upon these by formalizing tool use within a beam search framework and introducing an adaptive scoring mechanism to correct the selected actions.

## 3. Method

In this section, we first formalize multi-step visual reasoning as a dynamic Markov decision process (MDP). We then present the V-ABS framework, which implements the reasoning process through the thinker, actor, and observer. Subsequently, we describe the adaptive scoring mechanism for inference and the SFT strategy for training.

### 3.1. Problem Formulation

Standard MLLM inference typically maximizes the likelihood of a text sequence $Y$ conditioned on a static image $I$, formulated as $\max \pi_\theta(Y|I)$. We argue that for complex tasks, the optimal solution $Y^*$ depends on intermediate visual states that must actively evolve from $I$. Therefore, we formulate visual reasoning as a sequential decision-making process within a partially observable MDP tuple $(\mathcal{S}, \mathcal{A}, \mathcal{T}_{tool}, \mathcal{E})$. The state space $\mathcal{S}$ consists of states $s_t = (v_t, h_t)$ comprising the evolving visual context $v_t$ initialized at $v_0 = I$ along with the interaction history $h_t$. The set $\mathcal{A}$ indicates the action space. The transition function $\mathcal{T}_{tool}$ defines the non-differentiable update of visual state $s_t$ based on action $a_t$, formulated as $\mathcal{T}(s_t, a_t) \rightarrow s_{t+1}$. The process concludes based on a termination condition $\mathcal{E}$ defined by reaching a goal state or the max depth limitation $D$. To solve this MDP, we employ beam search (Wiseman & Rush, 2016) to approximate the optimal trajectory. The transition to the next nodes set $\mathcal{B}_{t+1}$ in beam search involves expanding the state space from current nodes $\mathcal{B}_t$ and sampling the top-$K$ trajectories with the highest likelihood:

$$\mathcal{B}_{t+1} = \underset{\{(s_t, a)|s_t \in \mathcal{B}_t, a \in \mathcal{A}\}}{\text{Top-K}} \left( \log \pi_\theta(Y^*|s_i, a) \right) \quad (1)$$

where $\pi_\theta$ represents the prior policy. However, relying solely on this prior usually leads to suboptimal performance, as it neglects the critical visual evidence revealed in the updated state $s_{t+1}$.

### 3.2. The V-ABS Framework

The core contribution of V-ABS lies in its architecture that integrates thinking, acting, and observing into a unified beam search process. Specifically, given the current state

**Algorithm 1** Visual Action-Observer Driven Beam Search

1:  **Input:** Image $I$, Beam Num $K$, Max Depth $D$
2:  **Output:** Optimal Node $n^*$
3:  $\mathcal{B}_0 \leftarrow \{(s_0 = (I, \emptyset), S_0 = 0)\}$;
4:  **for** $t = 0$ **to** $D - 1$ **do**
5:      $\mathcal{C} \leftarrow \emptyset$
6:      **for each** node $n$ in $\mathcal{B}_t$ **do**
7:          $(v_t, h_t) \leftarrow n.s$
8:          THINKER: $\mathcal{F}_{pri} \leftarrow \pi_\theta(s_t, \mathcal{A}_t)$
9:          $w_p \leftarrow \text{Ent}(\mathcal{F}_{pri})$
10:         **for** $b = 1$ **to** $B$ **do**
11:             ACTOR: $v_{t+1}^b \leftarrow \mathcal{T}_{\text{tool}}(v_t)$
12:             $s_{t+1}^b \leftarrow (v_{t+1}^b, h_t \oplus \text{Desc}(a_t^b))$
13:             OBSERVER: $\mathcal{F}_{obs}^b \leftarrow \pi_\theta(s_{t+1}^b)$
14:             HEURISTIC: $\mathcal{F}_{heur}^b \leftarrow \mathcal{H}(s_{t+1}^b)$
15:             $S_{t+1}^b \leftarrow AdaWeight(\mathcal{F}_{pri}, \mathcal{F}_{obs}, \mathcal{F}_{heur})$
16:             $\mathcal{C}.add((s_{t+1}^b, S_{t+1}^b))$
17:         **end for**
18:     **end for**
19:     $\mathcal{B}_{t+1} \leftarrow \text{TopK}(\mathcal{C}, K)$
20:     **if** $\mathcal{E}(\text{Best}(\mathcal{B}_{t+1}))$ **then**
21:         **break**
22:     **end if**
23: **end for**
24: **Return** $n^*$ from Best($\mathcal{B}$)

$s_t = (v_t, h_t)$ in each step, the MLLMs thinker module $\pi_\theta$ first evaluates the potential utility of candidate actions $\mathcal{A}_t = \{a_t^1, \ldots, a_t^B\}$, where $B$ is the total number of actions. To derive a robust confidence metric, we aggregate the prediction probabilities across a set of pre-defined positive attribute tokens $\mathcal{W}_{pos}$, which includes tokens such as "Yes", "True", and "Correct". The expectation $E_{pri}^b$ for the $b$-th action is calculated as the summed log-likelihood of these tokens, formulated as:

$$E_{pri}^b = \sum_{w \in \mathcal{W}_{pos}} \log \pi_\theta(w | s_t, a_t^b) \quad (2)$$

We then normalize all these expectations across the candidate action set to obtain the prior score $\mathcal{F}_{pri}$ via a temperature-scaled softmax function:

$$\mathcal{F}_{pri}^b = P(a_t^b | s_t) = \frac{\exp(E_{pri}^b / \tau)}{\sum_{j=1}^B \exp(E_{pri}^j / \tau)} \quad (3)$$

where $\tau = 0.5$ is a temperature hyperparameter. We employ the parallel inference strategy to compute prior scores of the entire action space $\mathcal{A}_t$ simultaneously, which ensures computational efficiency.

Then, the actor module executes the candidate actions by invoking the external tool function $\mathcal{T}_{\text{tool}}$ to transition the

visual context from $v_t$ to $v_{t+1}$. Meanwhile, the structured textual description of the executed action is integrated into the interaction history $h_t$. The state update is formulated as:

$$v_{t+1}^b = \mathcal{T}_{tool}(v_t, a_t^b; \phi), \quad h_{t+1}^b = h_t \oplus \text{Desc}(a_t^b), \quad (4)$$

where $\phi$ denotes the tool-specific parameters and $\text{Desc}(\cdot)$ generates the description for each action. The updated state $s_{t+1}^b = (v_{t+1}^b, h_{t+1}^b)$ provides grounded visual evidence for the following process.

Subsequently, the observer module serves as the critical feedback component that distinguishes V-ABS from previous approaches. We feed the updated states $s_{t+1}^b$ into the MLLMs and query the model regarding the correctness of the new visual context. Similar to the thinker, we derive the raw observation score $E_{obs}^b$ by aggregating the logits of the positive verification tokens $\mathcal{W}_{pos}$:

$$E_{obs}^b = \sum_{w \in \mathcal{W}_{pos}} \log \pi_\theta(w | s_{t+1}^b) \quad (5)$$

These raw scores are then normalized via softmax to yield the observation score $\mathcal{F}_{obs} \in [0, 1]^B$, formulated as:

$$\mathcal{F}_{obs}^b = \frac{\exp(E_{obs}^b)}{\sum_{j=1}^B \exp(E_{obs}^j)} \quad (6)$$

The thinker-actor-observer sequence forms one complete closed-loop in each step. After computing both prior and observer scores, the framework applies the adaptive weighting mechanism described in Section 3.3 and aggregates them with the task-specific heuristic score $\mathcal{F}_{heur}$ to determine the final score for each candidate. Then, we select the top-$K$ states based on the final scores and proceed to the next step. This mechanism effectively identifies cases where the thinker exhibits high uncertainty but the resulting visual state enables the facilitation of correct reasoning, thereby mitigating the influence of IAO bias. The detailed workflow is presented in Algorithm 1.

### 3.3. Adaptive Weighting Mechanism

**Entropy-based weights.** To effectively mitigate the IAO bias, it is reasonable to fuse the confidence from the thinker and observer dynamically. Specifically, we first quantify the thinker's uncertainty by calculating the Shannon entropy of the prior action probability distribution. Given the entropy $H_t = -\sum_b P(a_t^b | s_t) \log P(a_t^b | s_t)$, we define the adaptive weight $w_p$ for the prior as a sigmoid function of the entropy:

$$w_p = \frac{1}{1 + e^{\beta(H_t - \mu)}}, \quad w_o = 1 - w_p \quad (7)$$

where $\beta = 2$ controls the sensitivity and $\mu = 0.5$ is the uncertainty threshold. $w_o$ defines the weight of observer scores. When the thinker exhibits high entropy, we have

$w_p \to 0$ and $w_o \to 1$, forcing the algorithm to rely on the observer score $\mathcal{F}_{obs}$. Conversely, when the entropy is low, we have $w_p \to 1$. To enhance inference efficiency, we introduce an acceleration mechanism: when the entropy falls below a strict threshold $\delta$, indicating high confidence in the prior, we bypass the computationally expensive observer module and directly set $w_o = 0$.

**Theoretical analysis.** Our adaptive mechanism is motivated by statistical estimation theory. Let $Q^*$ be the true utility of an action. The thinker provides a prior estimate $Q_{pri} \sim \mathcal{N}(Q^*, \sigma_{pri}^2)$, where the variance $\sigma_{pri}^2$ reflects the uncertainty of the prior estimation. The observer provides a posterior measurement $Q_{obs} \sim \mathcal{N}(Q^*, \sigma_{obs}^2)$, which is grounded in actual pixels and assumed to have lower, stable variance. We seek a combined estimator $\hat{Q} = w_p Q_{pri} + (1 - w_p) Q_{obs}$ that minimizes the mean squared error $\mathbb{E}[(\hat{Q} - Q^*)^2]$. According to the inverse variance weighting principle, the optimal weight $w_p^*$ is determined by the precision ratio:

$$w_p^* = \frac{\sigma_{obs}^2}{\sigma_{pri}^2 + \sigma_{obs}^2} = \frac{1}{1 + \frac{\sigma_{pri}^2}{\sigma_{obs}^2}} \quad (8)$$

Eq. 8 reveals that the weight of the thinker should decay as its variance relative to the observer increases. Our entropy-based formula serves as a differentiable approximation of this theoretical optimum. By mapping the thinker's variance to entropy as $\sigma_{pri}^2 \propto e^{\beta H_t}$ and treating the observer's variance as a normalizing constant, our sigmoid function $w_p(t) \approx (1 + e^{\beta H_t})^{-1}$ effectively mimics the optimal weighting behavior. Thus, V-ABS dynamically minimizes estimation error by combining the prior predictions and grounded observations.

**Final scoring formula.** The total score $S_{t+1}^b$ for a state transition $s_{t+1}^b$ is the weighted sum of the adaptive components, formulated as:

$$S_{t+1}^b = w_p \cdot \mathcal{F}_{pri}^b + \mathbb{I}(H_t \geq \delta) \cdot w_o \cdot \mathcal{F}_{obs}^b + \mathcal{F}_{heur}^b \quad (9)$$

where $\mathbb{I}(\cdot)$ is the indicator function for the acceleration mechanism, and $\mathcal{F}_{heur}^b$ represents task-specific heuristic scores based on the updated state $s_{t+1}^b$, such as penalizing states that are far from the goal. This formula is applied at the end of each iteration to rank all candidate states, followed by selecting the top-$K$ states with the highest scores to form the beam set $\mathcal{B}_{t+1}$ for the next iteration. Finally, we will end the search algorithm if the best node in $\mathcal{B}_{t+1}$ satisfies the termination condition $\mathcal{E}$, such as reaching the maximum depth $D$ or exceeding a predefined score threshold.

### 3.4. SFT for Reducing Prior Uncertainty

Complementing inference-time adaptation, we employ supervised fine-tuning (SFT) to structurally mitigate the in-

trinsic prior uncertainty $\sigma_{pri}^2$ within the MLLM. Diverging from standard long-horizon generation, our approach utilizes a dataset of $\sim$80k samples to focus exclusively on subsequent action verification. Formally, for a given state $s_t$, we partition the candidate action space $\mathcal{A}_t$ into a subset of optimal actions $\mathcal{A}_t^r$ and erroneous ones $\mathcal{A}_t^e = \mathcal{A}_t \setminus \mathcal{A}_t^r$. We fine-tune the model to predict a binary verification token $y \in \{\text{``Yes''}, \text{``No''}\}$ conditioned on a sampled action $a$, minimizing the following objective:

$$\mathcal{L}_{\text{SFT}} = -\mathbb{E}_{(s_t,a)\sim\mathcal{D}} \Big[ \mathbb{I}(a \in \mathcal{A}_t^r) \log \pi_\theta(\text{``Yes''}|s_t, a)$$
$$+ \mathbb{I}(a \in \mathcal{A}_t^e) \log \pi_\theta(\text{``No''}|s_t, a) \Big] \quad (10)$$

By explicitly maximizing the likelihood of positive indicators for correct actions in the training, the model enables to minimize the prior entropy in the inference stage and leads to an optimal search path.

## 4. Experiments

### 4.1. Experimental Setup

**Implementation Details.** We adopt Qwen3-VL-8B (Yang et al., 2025), Qwen2.5-VL-7B (Wang et al., 2024), and Intern-VL3-8B (Wang et al., 2025b) as our open-source baselines. For proprietary model comparison, we employ on GPT-4o (Hurst et al., 2024). The supervised fine-tuning (SFT) is conducted on $16 \times$ NVIDIA H100-80G GPUs for 200 steps, using a batch size of 128 and a learning rate of $1 \times 10^{-5}$. During inference, we utilize a single GPU, setting the beam size to $K = 3$ and employing dynamic search depth $D$ for different tasks, unless otherwise specified.

**Benchmarks and Tasks.** We evaluate V-ABS on three distinct categories of tasks to demonstrate the robustness and generalization:

- Fine-grained Visual Search: We use V* (Wu & Xie, 2024) to assess attribute recognition and spatial relationship understanding. Additionally, we employ HR-Bench-4K/8K (Wang et al., 2024) to evaluate perception in ultra-high-resolution scenarios. We utilize the `Image.Crop` function as the $\mathcal{T}_{tool}$ in this task.

- Visual Navigation: We validate in the VisuoThink (Wang et al., 2025c) (Levels 3-5) and Frozen Lake ($4\times4$ to $8\times8$ grids) like VSP (Wu et al., 2025) to test sequential decision-making. We include TIR-Bench Maze (Li et al., 2025a) for long-horizon path planning. We also utilize the `Image.Crop` function to obtain the regional views in different directions.

- Visual Manipulation & Logic: We evaluate on Jigsaw task like (Li et al., 2025a) to test regional reasoning and Sudoku (Ghosal et al., 2025) with $9 \times 9$ grids for logical constraint satisfaction. The tool functions in these two tasks are defined as `Image.Rearrange` and `Image.Filling`. The details of above tool functions are described in Appendix C.

## 4.2. Main Results: Performance on the V-ABS Pipeline

We initiate our evaluation by benchmarking V-ABS in a training-free setting against state-of-the-art methods. As evidenced in Table 1, V-ABS establishes dominance in high-resolution visual search. When equipped with V-ABS, Qwen2.5-VL achieves overall scores of 91.1%, 79.0%, and 77.5% on V*, HR-Bench-4K, and HR-Bench-8K, respectively, yielding substantial improvements of 15.2%, 13.5%, and 15.5% over the base model. Furthermore, the consistent gains observed across diverse backbones, including Qwen3-VL (Yang et al., 2025), Intern-VL3 (Wang et al., 2025b), and GPT-4o (Hurst et al., 2024), underscore the framework's universality across both open-source and proprietary models. Crucially, V-ABS outperforms previous advanced methods such as ZoomEye (Shen et al., 2025), DeepEyes V2 (Hong et al., 2025), and Pixel-Reasoner (Wang et al., 2025a). We attribute this superiority to our active verification mechanism. While approaches like ZoomEye rely on heuristic zooming, they lack feedback to validate the utility of the zoomed region. In contrast, V-ABS's adaptive weighting mechanism filters out uninformative visual states, effectively mitigating the IAO bias.

Table 2 highlights the efficacy of our algorithm in the navigation task. Specifically, V-ABS executes robust multi-step planning, boosting the accuracy of Qwen3-VL by 29.9%, 46.8%, and 30.6% across various difficulty levels in VisuoThink-Nav (Wang et al., 2025c). Similarly, on the Frozen-Lake and TIR-Bench Maze tasks, V-ABS enables Qwen3-VL to achieve average success rates of 30.0% and 65.0%, respectively, vastly outperforming the baseline. When compared against previous tree search methods (Wang et al., 2025c) conducted on GPT-4o, V-ABS also demonstrates superior navigational capabilities. This confirms that active visual operations and the observation of potential collisions allow the model to proactively prune fatal trajectories.

Expect the aforementioned tasks operate within discrete and pre-defined action spaces, like cropping quadrants for search or cardinal directions for navigation, we further validate V-ABS on open-ended manipulation tasks to assess generalization. As presented in Table 3, our method predicts and scores candidates from the open actions. On the Jigsaw (Li et al., 2025a) and Sudoku (Ghosal et al., 2025) tasks, V-ABS yields improvements of 5.0% and 23.4% over the Qwen3-VL-8B baseline, respectively. These consis-

tent gains observed across other models demonstrate that V-ABS accommodates both pre-defined and open-ended action spaces, ensuring broad applicability and robustness in diverse visual reasoning scenarios.

## 4.3. SFT Enhancement and Action Space Analysis

We investigate the interplay between SFT and action space complexity in Table 4. For tasks with restricted action spaces, such as V* with quadrant zooming or Frozen Lake with 4-way movement, the training-free V-ABS alone contributes the vast majority of performance gains, improving from 75.9% to 90.5% in V*. Here, the search space is small enough for the observer to efficiently filter candidates even with a weak prior. SFT adds only marginal gains (+0.6% on V*) by sharpening the prior policy, indicating that inference-time verification is the primary driver for discrete decision-making. In contrast, for open-ended action like Jigsaw, V-ABS alone yields negligible improvement (5.3% → 7.8%). The vastness of the open action space renders unguided search ineffective. SFT is critical here, boosting performance to 81.2%. It operates as a crucial regularization mechanism that effectively constrains the policy distribution to a valid manifold, thereby ensuring that the beam search focuses on plausible trajectories rather than diverging into irrelevant spaces. Consequently, we conclude that while the training-free V-ABS framework is sufficient for addressing discrete reasoning tasks, SFT remains an indispensable prerequisite for achieving robust performance in open-ended manipulation.

## 4.4. Search Strategy Comparison

We compare V-ABS against standard test-time scaling methods, specifically beam search (Snell et al., 2024) and Monte Carlo tree search (MCTS) (Feng et al., 2023). As shown in Table 5, applying these methods to Qwen2.5-VL yields notable gains and improves the accuracy in HR-Bench-8K from 62.0% to 71.0%. However, their performance eventually saturates. The fundamental limitation is that these methods select search paths based solely on the model's internal reasoning (i.e., the prior score), without verifying the actual visual outcome of the executed actions. Consequently, they remain susceptible to the IAO bias, where the model incorrectly predicts the utility of an action. V-ABS overcomes this by incorporating the observer score, which evaluates the updated visual state to verify the reasoning path. Achieving 77.5% on HR-Bench-8K and 91.1% on V*, our results demonstrate that effective visual reasoning requires not only planning actions but also validating them against physical visual feedback.

## 4.5. Ablation Study

We conduct ablation studies to analyze the internal mechanisms of V-ABS, focusing on three key aspects: search

*Table 1.* Performance comparison on fine-grained visual search tasks. We evaluate on V* (Attribute, Spatial, Overall) and HR-Bench at 4K and 8K resolutions. Rows highlighted in gray denote our proposed V-ABS method. The values in *green italics* indicate the absolute improvement points over the corresponding base model. The best performance are highlighted in **bold**.

| Method | V* Benchmark | | | HR-Bench-4K | | | HR-Bench-8K | | |
|---|---|---|---|---|---|---|---|---|---|
| | Attr | Spatial | Overall | FSP | FCP | Overall | FSP | FCP | Overall |
| LLaVA-v1.6-13B | 60.0 | 64.5 | 61.8 | 49.8 | 41.3 | 45.5 | 38.0 | 38.3 | 38.1 |
| LLaVA-HR-X-7B | 51.3 | 64.5 | 56.5 | 57.8 | 46.3 | 52.0 | 42.0 | 41.3 | 41.6 |
| Qwen2.5-VL-7B | 76.5 | 75.0 | 75.9 | 83.0 | 48.0 | 65.5 | 81.0 | 43.0 | 62.0 |
| *Previous Methods* | | | | | | | | | |
| Qwen2.5-VL + Thyme | 83.5 | 80.3 | 82.2 | **91.0** | 63.0 | 77.0 | 86.5 | 57.5 | 72.0 |
| Qwen2.5-VL + DeepEyes v2 | - | - | 81.8 | - | - | 77.9 | - | - | 73.8 |
| Qwen2.5-VL + ZoomEye | 88.7 | **89.4** | 89.0 | 86.8 | 53.5 | 70.1 | 84.8 | 52.0 | 68.4 |
| Qwen2.5-VL + Pixel-Reasoner | - | - | 84.3 | - | - | 74.0 | - | - | 66.9 |
| Qwen2.5-VL + V-ABS | **93.0** | 88.2 | **91.1** | **91.0** | 67.0 | 79.0 | 89.0 | 66.0 | **77.5** |
| *Other Baselines* | | | | | | | | | |
| Qwen3-VL | 73.0 | 80.3 | 75.9 | 85.0 | 55.0 | 70.0 | 82.0 | 55.0 | 68.5 |
| + V-ABS | **91.2** *+18.2* | 89.3 *+9.1* | 90.5 *+14.6* | 90.0 *+5.0* | 62.0 *+7.0* | 76.0 *+6.0* | 88.0 *+6.0* | 68.0 *+13.0* | 78.0 *+9.5* |
| Intern-VL3 | 73.0 | 72.4 | 72.8 | 76.0 | 63.0 | 69.5 | 61.0 | 60.0 | 60.5 |
| + V-ABS | 90.4 *+17.4* | 79.0 *+6.6* | 85.9 *+13.1* | 86.0 *+10.0* | 65.0 *+2.0* | 75.5 *+6.0* | 72.0 *+11.0* | 60.0 *+0.0* | 66.0 *+5.5* |
| GPT-4o | 67.0 | **80.3** | 72.3 | 70.0 | 48.0 | 59.0 | 49.0 | 67.0 | 58.0 |
| + V-ABS | 83.5 *+16.5* | 67.1 | **77.0** *+4.7* | 80.8 *+10.8* | 62.0 *+14.0* | 71.4 *+12.4* | 74.8 *+25.8* | 54.1 | **64.5** *+6.5* |

*Table 2.* Performance on visual navigation benchmarks, including VisuoThink (Levels 3-5), Frozen-Lake (grid sizes from $4 \times 4$ to $8 \times 8$), and TIR-Bench-Maze. V-ABS significantly outperforms baselines in the long-sequence tasks, demonstrating remarkable robustness and planning capabilities.

| Method | VisuoThink (Level) | | | Frozen-Lake (Grid) | | | TIR-Bench |
|---|---|---|---|---|---|---|---|
| | L-3 | L-4 | L-5 | 4×4 | 6×6 | 8×8 | Maze |
| Qwen3-VL-8B | 46.9 | 25.9 | 19.0 | 25.0 | 15.0 | 2.5 | 17.5 |
| Qwen2.5-VL-7B | 18.2 | 14.2 | 14.7 | 15.0 | 10.5 | 2.5 | 10.7 |
| Intern-VL3-8B | 29.0 | 20.5 | 7.4 | 10.0 | 12.5 | 5.0 | 26.7 |
| Qwen3-VL + V-ABS | **76.8** *+29.9* | **72.7** *+46.8* | **49.6** *+30.6* | **55.0** *+30.0* | **22.5** *+7.5* | **12.5** *+10.0* | **65.0** *+47.5* |
| Qwen2.5-VL + V-ABS | 34.8 *+16.6* | 23.7 *+9.5* | 18.2 *+3.5* | 45.0 *+30.0* | 12.5 *+2.0* | 10.0 *+7.5* | 46.7 *+36.0* |
| Intern-VL3-8B + V-ABS | 51.5 *+22.5* | 40.5 *+19.9* | 26.8 *+19.4* | 30.0 *+20.0* | 15.0 *+2.5* | 10.0 *+5.0* | 56.7 *+30.0* |
| GPT-4o | 42.9 | 22.9 | 22.9 | 20.0 | 15.0 | 5.0 | 17.5 |
| GPT-4o + VisuoThink | 93.8 | 61.0 | 49.0 | 35.0 | 20.0 | **7.5** | 20.1 |
| GPT-4o + V-ABS | **94.9** *+52.0* | **74.1** *+51.2* | **69.7** *+46.8* | **50.0** *+30.0* | **25.0** *+10.0* | 7.5 *+2.5* | **23.3** *+5.8* |

*Table 3.* Performance on visual manipulation tasks (Jigsaw and Sudoku). V-ABS consistently enhances accuracy across diverse backbones by leveraging dynamic visual verification to solve geometric constraints.

| Method | Jigsaw | | | Sudoku |
|---|---|---|---|---|
| | Res3 | Res4 | Res5 | Acc |
| Qwen3-VL | 10.0 | 5.3 | 2.7 | 14.4 |
| + V-ABS | **21.0** *+11.0* | **7.8** *+2.5* | **4.3** *+1.6* | **37.8** *+23.4* |
| Qwen2.5-VL | 8.5 | 5.0 | 2.6 | 14.0 |
| + V-ABS | 19.2 *+10.7* | 11.2 *+6.2* | 3.2 *+0.6* | 32.5 *+18.5* |
| Intern-VL3 | 9.8 | 5.4 | 2.7 | 11.3 |
| + V-ABS | 18.4 *+8.6* | 11.3 *+5.9* | 3.2 *+0.5* | 33.1 *+21.8* |

*Table 4.* Impact of SFT on Qwen3-VL-8B baseline. SFT provides marginal gains on tasks with restricted action spaces (V*, Frozen-Lake). However, it is critical for open-ended action tasks to effectively constrain the vast search space.

| Configuration | Close Action | | Open Action |
|---|---|---|---|
| | V* | Frozen-Lake | Jigsaw (Res4) |
| Raw | 75.9 | 17.5 | 5.3 |
| + V-ABS | 90.5 | 35.0 | 7.8 |
| + V-ABS + SFT | **91.1** | **42.5** | **81.2** |

depth, the contribution of scoring components, and the effect of SFT on model uncertainty.

**Trade-off Between Accuracy and Cost.** We first analyze the trade-off between reasoning accuracy and computational

efficiency across max search depths $D$ ranging from 1 to 16 in the visual search task V*. As illustrated in Figure 3(a), the accuracy achieves a peak at depth 3. The performance ascent from depth 1 to 3 confirms that multi-step verification is essential for resolving visual ambiguities. However, accuracy slightly degrades beyond this optimal depth, suggesting that overly long trajectories introduce reasoning noise. The

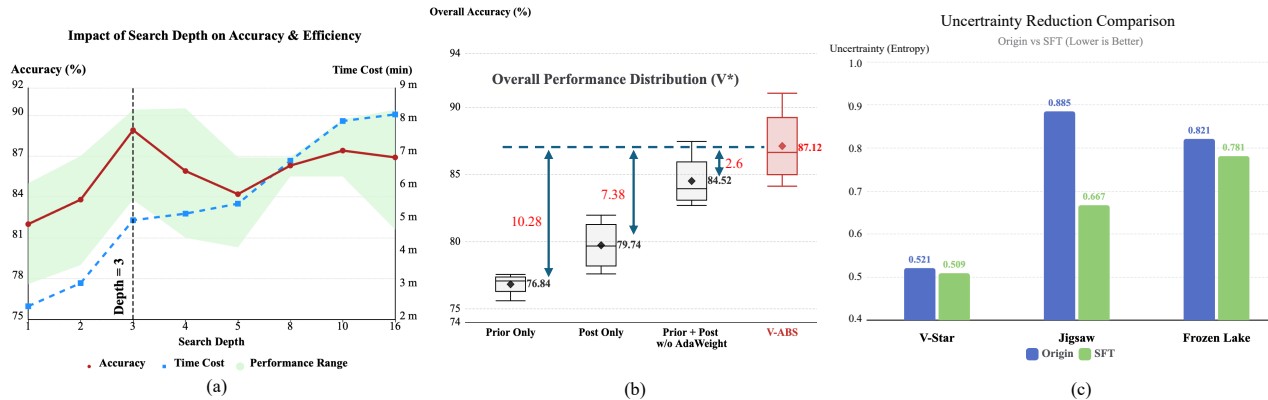

*Figure 3.* Ablation Studies. (a) Trade-off between accuracy and cost: accuracy peaks at max depth $D = 3$ while time cost scales linearly. The green band indicates the performance range. (b) Component analysis: Relying solely on prior or post scores leads to significant drops, while the adaptive weighting contributes more gains than direct summation. (c) Impact of SFT: SFT reduces the uncertainty across all tasks, with the most significant discrepancy observed in open-ended tasks like Jigsaw.

*Table 5.* Ablation study of Search Strategies on Qwen models. V-ABS consistently outperforms previoussearch methods (Beam/MCTS) by leveraging active visual updates.

| Strategy | V* | HR-4K | HR-8K |
|---|---|---|---|
| *MLLM: Qwen3-VL-8B* | | | |
| Base Model | 75.9 | 70.0 | 68.5 |
| + Beam Search | 84.3 | 71.0 | 75.0 |
| + MCTS | 80.1 | 70.0 | 70.5 |
| **+ V-ABS (Ours)** | **90.5** | **76.0** | **78.0** |
| *MLLM: Qwen2.5-VL-7B* | | | |
| Base Model | 75.9 | 65.5 | 62.0 |
| + Beam Search | 81.2 | 74.0 | 70.5 |
| + MCTS | 80.1 | 71.0 | 71.0 |
| **+ V-ABS (Ours)** | **91.1** | **79.0** | **77.5** |

widening green band defines the performance range, indicating increased instability at higher depths. Simultaneously, the time cost scales linearly, validating the efficiency of the beam search strategy and demonstrating that deep search remains computationally feasible.

**Ablation of the Adaptive Weighting Components.** As shown in Figure 3(b), we validate the effectiveness of each component in our adaptive weighting mechanism in the V* benchmark. Specifically, relying solely on the prior scores $\mathcal{F}_{pri}$ leads to the most severe performance degradation with a drop of 10.28%, proving that internal imagination alone creates a hallucination bubble without visual grounding. Conversely, depending solely on the observer scores $\mathcal{F}_{obs}$ leads to a 7.38% decline. Furthermore, ablating the adaptive mechanism in favor of static averaging results in a 2.6% decrease, confirming that dynamically modulating weights based on entropy yields a more reasonable fusion strategy than rigid aggregation.

**Impact of SFT on the Prior Uncertainty.** Finally, we investigate the mechanism underlying SFT by analyzing the uncertainty of the action probability distribution. As illustrated in Figure 3(c), the impact of SFT correlates strongly with the complexity of the action space. For open-ended tasks like Jigsaw, SFT significantly reduces the uncertainty from 88.5% to 66.7%, effectively sharpening the prior policy to narrow the action space. This reduction in uncertainty directly accounts for the substantial performance gains observed in the main experiments. In contrast, for tasks with restricted action spaces, such as cropping in V*, the reduction in uncertainty is negligible (from 52.1% to 50.9%). This corroborates our finding that the base model already possesses sufficient confidence within low-dimensional discrete spaces, limiting the marginal utility of additional SFT.

## 5. Conclusion

In this work, we presented V-ABS, an agentic inference framework that transitions MLLMs from static perception to dynamic, action-driven exploration. By integrating tool-mediated state updates within a closed-loop beam search, V-ABS grounds the reasoning process in physical visual evidence, effectively overcoming the limitations of static textual priors. We further uncovered the imagination-action-observer (IAO) bias and proposed an entropy-based adaptive weighting algorithm to mitigate it by dynamically calibrating internal planning confidence with external visual feedback. Extensive experiments demonstrate that V-ABS achieves state-of-the-art performance across diverse visual reasoning benchmarks. Moreover, our analysis reveals a critical mechanism-dependent insight: while training-free search is largely sufficient for tasks with restricted action spaces, SFT is indispensable for open-ended manipulation tasks by effectively narrowing the search space. Future

work will explore optimizing this dynamic interaction loop through reinforcement learning, moving towards models that inherently align their imagination with visual reality.

## Acknowledgement

This work is partially supported by National Natural Science Foundation of China (Grant No. 62402318, 62376153, 24Z990200676), AI for Science Seed Program of Shanghai Jiao Tong University (Grant No. WH410261901/002), Shanghai Municipal Special Program for Basic Research on General AI Foundation Models (Grant No. 2025SHZDZX025G12).

## Impact Statement

This paper presents work whose goal is to advance the field of Machine Learning. There are many potential societal consequences of our work, none which we feel must be specifically highlighted here.

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

# A. Details of Evaluation Datasets

We evaluate V-ABS across three categories of visual reasoning tasks, including fine-grained visual search, visual navigation, and visual logic. The detailed statistics and metrics are summarized below.

## A.1. Fine-grained Visual Search.

**V\* Benchmark.**    (Wu & Xie, 2024) This benchmark features multiple-choice questions designed to assess the retrieval of small targets within high-resolution images. The dataset comprises 191 samples, consisting of 115 attribute-focused queries (e.g., object color or material) and 76 spatial relationship queries (e.g., relative positioning). We report the accuracy of the final answer derived from the visual search process.

**HR-Bench-4K/8K.**    (Wang et al., 2024) Designed to evaluate MLLMs on ultra-high-resolution images (4K and 8K), this benchmark focuses on Fine-grained single-instance perception (FSP) and cross-instance perception (FCP). It contains 200 images for each resolution setting, covering diverse tasks such as OCR, object counting, and attribute recognition.

## A.2. Visual Navigation.

**VisuoThink.**    (Wang et al., 2025c) This benchmark tasks agents with navigating a grid map from a start position *home* to a goal *office* while avoiding obstacles. The difficulty levels (e.g., 3, 4, 5) correspond to the number of turns required in the optimal path. Crucially, there exists a unique valid trajectory from the initial position to the goal.

**Frozen Lake.**    Drawing inspiration from the VSP benchmark (Wu et al., 2025), we synthesize a suite of Frozen Lake environments comprising humans, gifts, ice surfaces, and holes. Distinguishing our setup from prior works, we ensure the existence of a unique solvable path from the start (human) to the goal (gift). The dataset consists of 20, 40, and 40 samples across $4 \times 4$, $6 \times 6$, and $8 \times 8$ grid resolutions, respectively.

**TIR-Bench-Maze.**    (Li et al., 2025a) A standard benchmark for long-horizon path planning, consisting of 120 maze maps with varying resolutions. Agents must navigate from a top-left starting position to a bottom-right goal. The complexity is heightened by the presence of confusing crosses along the optimal path, which challenges the model's ability to maintain spatial consistency.

## A.3. Visual Logic and Manipulation.

**Jigsaw Puzzle.**    Adhering to the protocol in (Li et al., 2025a), we construct this task using images from the Ref-COCO dataset. Each image is partitioned into a grid of $N \times N$ patches, with $N \in \{3, 4, 5\}$ (denoted as Res3, Res4, and Res5). These patches are randomly shuffled and indexed from 1 to $N^2$. The model is tasked with predicting the correct permutation sequence to restore the original visual coherence.

**Sudoku.**    Following the protocol in (Ghosal et al., 2025), we construct a $9 \times 9$ visual Sudoku dataset. The model is tasked with filling in missing numbers according to standard Sudoku constraints. Crucially, the model is restricted to deriving information exclusively from the raw image, without access to structured text representations. This constraint ensures the rigorous validation of the model's visual reasoning capabilities rather than pure symbolic logic.

# B. SFT Dataset Construction

To bolster the verification capability of the Observer module and structurally reduce the entropy of the Thinker, we curated a specialized SFT dataset comprising approximately 80k samples. To ensure validity and effective transfer, the distribution of the training data is strategically aligned with the domain characteristics of our evaluation benchmarks, specifically targeting fine-grained visual search (V\*), visual navigation (Frozen-Lake), and visual logic (Jigsaw). The dataset is constructed as binary verification pairs $(s_t, a_t, y)$, strictly balancing positive samples (where action $a_t$ is correct, $y =$ "Yes") and negative samples (where $a_t$ leads to errors, $y =$ "No") to prevent label bias.

The detailed distribution is presented in Table 6. (1) Visual Search: We construct 7,040 samples similar to the V\* benchmark. This subset focuses on verifying whether a specific crop action accurately locates the target object described in the query. (2) Visual Navigation: We generate 14,000 samples to facilitate the capability of frozen-lake mapping. This includes 4,000 samples for basic state perception and 10,000 samples for verifying spatial movement actions like "If moving *Right* leads to a safe path?". (3) Visual Logic: Given the high complexity of open-ended manipulation, we allocate a large portion to the Jigsaw task. We uniformly distribute these right or error actions across $3 \times 3$, $4 \times 4$, and $5 \times 5$ resolutions to teach the

*Table 6.* Distribution of the SFT Dataset. We recalculate the data across three core domains to ensure the generalization, with a higher weight on complex visual tasks.

| Category | Data Type | Samples |
|---|---|---|
| Visual Search | Crop action (Correct Region) | 3,520 |
| | Crop action (Error Region) | 3,520 |
| Visual Nav. | Perception QA | 4,000 |
| | Spatial Action (Correct Direction) | 5,000 |
| | Spatial Action (Error Direction) | 5,000 |
| Visual Logic | Jigsaw $3 \times 3$ (Correct/Error) | 10k / 10k |
| | Jigsaw $4 \times 4$ (Correct/Error) | 10k / 10k |
| | Jigsaw $5 \times 5$ (Correct/Error) | 10k / 10k |
| **Total** | | **80,940** |

model robust geometric matching and permutation verification.

# C. Tool Definitions

We formally define the specific tool functions $\mathcal{T}_{tool}$ utilized within the actor module for each task domain.

**<Image.Crop> for Visual Search.**   We utilize the `Image.Crop` function to enable the model to focus on specific regions of interest. We define four candidate quadrants: *top-left, top-right, bottom-left,* and *bottom-right*. To prevent critical information from being truncated at the boundaries, the cropped region is set to cover $0.7 \times 0.7$ of the current image dimensions, ensuring sufficient overlap. The function takes the current image $v_t$ and a target directional parameter as input, outputting the zoomed-in sub-region $v_{t+1}$.

**<Image.Crop> for Visual Navigation.**   We adapt the cropping mechanism for grid-based navigation tasks. Distinct from the overlapping strategy in visual search, this variant strictly crops the specific grid cell corresponding to the agent's intended movement. The input includes the global map image and the target grid index. The function returns the local view of the target region. Simultaneously, the global map state is updated to reflect the agent's new current location.

**<Image.Rearrange> for Jigsaw.**   To manipulate the shuffled patches in the Jigsaw task, we define the `Image.Rearrange` function to recover the correct spatial arrangement. Specifically, the function takes three inputs: the current visual state $v_t$, the current patch permutation $P_{curr} = \{p_1, \ldots, p_{N^2}\}$, and the target permutation indices $P_{pred}$ predicted by the MLLMs. It then physically rearranges the patches from their current positions to the target indices, returning the reassembled image $v_{t+1}$.

**<Image.Filling> for Sudoku.**   The `Image.Filling` function is designed to fill in the missing digits within the Sudoku grid. It receives a set of target coordinates $\{(x_1, y_1), \ldots, (x_M, y_M)\}$ and the corresponding target values $\{val_1, \ldots, val_M\}$ predicted by the MLLMs. The function renders the digits onto the specified coordinates of the current grid and returns the updated Sudoku map $v_{t+1}$ for subsequent verification.

# D. Justification of Variance-Entropy Mapping.

We provide a rigorous derivation to justify the mapping $\sigma_{pri}^2 \propto e^{\beta H_t}$. Modeling the thinker's prior estimation as a Gaussian distribution $Q_{pri} \sim \mathcal{N}(\mu, \sigma_{pri}^2)$, its uncertainty is quantified by the differential entropy. Assuming the natural logarithm for

entropy calculation:

$$
\begin{aligned}
H(Q_{pri}) &= -\int_{-\infty}^{\infty} p(x) \log p(x)\, dx \\
&= -\int_{-\infty}^{\infty} p(x) \left[ -\frac{1}{2} \log(2\pi\sigma_{pri}^2) - \frac{(x-\mu)^2}{2\sigma_{pri}^2} \right] dx \\
&= \frac{1}{2} \log(2\pi\sigma_{pri}^2) + \frac{1}{2\sigma_{pri}^2} \underbrace{\int_{-\infty}^{\infty} p(x)(x-\mu)^2\, dx}_{\text{Variance } \sigma_{pri}^2} \\
&= \frac{1}{2} \log(2\pi\sigma_{pri}^2) + \frac{1}{2} = \frac{1}{2} \log(2\pi e \sigma_{pri}^2)
\end{aligned}
\tag{11}
$$

Rearranging to solve for the variance $\sigma_{pri}^2$:

$$
2H(Q_{pri}) = \log(2\pi e \sigma_{pri}^2) \implies e^{2H} \propto \sigma_{pri}^2
\tag{12}
$$

This derivation demonstrates that for a Gaussian belief state, the estimation variance is proportional to the exponential of its entropy ($\sigma^2 \propto e^{2H}$). This provides the theoretical grounding for our approximation $\sigma_{pri}^2 \propto e^{\beta H_t}$ (where $\beta \approx 2$), linking the information-theoretic uncertainty directly to the estimation variance.

## E. Details of the Prompts

We provide the prompt examples used for the Thinker and Observer modules.

---

### Template 1: Visual Search Prompts

**Role:** Expert Visual Assistant / Visual Question Answering Agent

**[Thinker]** You are a visual assistant looking for an answer to Question: `<question>`
Focus your attention specifically on the [*Top-Right/Top-Left/Bottom-Right/Bottom-Left*] quadrant of the Image ``.
Does this specific region contain the object or visual clues needed to answer the question? Answer *Yes* or *No*.

---

**[Observer]** Question: `<question>`
Look at the current image view ``.
Does this image contain sufficient and clear visual evidence to answer the question correctly? Please answer **Yes** or **No**.

---

**[Final Answer]** You are an intelligent visual question answering agent. Question: `<question>` You have collected the following visual evidence paths: ``···`` Carefully analyze the evidence and answer the question. If it is a multiple-choice question, output the option letter.

---

### Template 2: Visual Navigation Prompts

**Role:** Navigator / Map Auditor

**[Thinker]** There is the map image ``. You are in the position of [*human/home*]. Considering moving [*Upper/Down/Left/Right*] to the next position.
Task:
Analyze if the Proposed Action is valid and logical.
- Is the direction BLOCKED by a wall immediately?
- Does it move drastically away from the goal?
Question: Is moving [*Upper/Down/Left/Right*] a valid and reasonable step to take right now? Please answer **Yes** or **No**.

---

**[Observer]** Action Execution Assessment:
Action Taken:

-[*Upper/Down/Left/Right*]
Images:
- Full Map (After Move): ``
- Regional View (Target Cell): ``
Task:
Verify the safety of the New Position. Look at the regional view. This shows the context of the target location.
- Is it a bad path (Black or Obstacle Area)?
- Or is it a safe or right path (White/Light Area)?
Question: Is the target position clearly a safe and correct road? Please answer **Yes** or **No**

## Template 3: Visual Manipulation Prompts (Jigsaw & Sudoku)

**Part A: Jigsaw Puzzle**
[**Thinker**] Current Image:
- ``.
You are an assistant to recover the correct order based on the current image. Please output at least `<cands>` candidate set of the right permutation, and do you think it contains high confidence, output **Yes** or **No**.
Output Format:

```
{
    "candidates": [
        { "permutation": [[1,2,3], ...], "High Confidence": "Yes/No" }
    ]
}
```

[**Observer**] Given the original image `` and the rearranged image ``
Do you think the arrangement process recovers the correct order based on the original image?
Please answer **Yes** or **No**.

---

**Part B: Sudoku**
[**Thinker**] Current Image:
-``
Task:
- Fill in the remaining positions with numbers (1-9) according to the rules of Sudoku, and correspond them with their row and column numbers in the image, while also providing the reasoning.
- Make sure there are no duplicate digits in each Column.
- Make sure there are no duplicate digits in each Row.
- Make sure there are no duplicate digits in each bold 3*3 Box.
Please provide at least `<cands>` candidates and your confidence with **Yes** or **No**.

```
{
    "candidates": [
        {
          "coordinates": [[1,1],...],
          "value": [5,...],
          "High Confidence": "Yes/No"
        },
        ...
    ]
}
```

[**Observer**] You are a Sudoku Validator.
Updated Sudoku map:
- ``
Task:

> - Verify the validity of the filled numbers (masked as blue) recently in the updated Sudoku Map:
>  Checklist for each number:
> - Is it unique in each Row?
> - Is it unique in each Column?
> - Is it unique in each bold 3x3 Box?
>  Please answer **Yes** or **No**

## F. Efficiency Analysis Across Search Depths

We conduct a systematic efficiency study on the VisuoThink benchmark to show how computational cost and accuracy jointly scale with search depth $D$. For a fair comparison, beam width $K$ and number of parallel workers are both fixed at 1; only $D$ is varied from 1 to 4. We also include two competitive baselines: standard Beam Search (fixed depth $D = 3$) and Monte-Carlo Tree Search (MCTS, $D = 3$), evaluated under the same single-worker setting.

*Table 7.* Efficiency study on VisuoThink (beam width $K = 1$, single worker). Time cost is per-sample wall-clock time without parallelisation.

| Benchmark | Method | Avg API Calls | Avg Tokens (K) | Time (s) | Accuracy |
|---|---|---|---|---|---|
| VisuoThink | V-ABS ($D = 1$) | 9.8 | 2.3 | 22 | 32.6% |
| VisuoThink | V-ABS ($D = 2$) | 15.2 | 3.7 | 28 | 41.7% |
| VisuoThink | V-ABS ($D = 3$) | 20.1 | 4.9 | 32 | 45.8% |
| VisuoThink | V-ABS ($D = 4$) | 23.2 | 5.2 | 37 | 46.1% |
| VisuoThink | Beam Search ($D = 3$) | 16.0 | 3.2 | 26 | 31.3% |
| VisuoThink | MCTS ($D = 3$) | 25.2 | 5.1 | 39 | 38.7% |

Table 7 shows that accuracy scales sub-linearly with $D$: the largest single-step gain occurs from $D = 1$ to $D = 2$ (+9.1%), while the marginal gain from $D = 3$ to $D = 4$ is only +0.3%. Compared with Beam Search and MCTS at $D = 3$, V-ABS achieves 45.8% accuracy with fewer API calls than MCTS (20.1 vs. 25.2) and a 14.5% absolute improvement over Beam Search (31.3%), demonstrating that the thinker-actor-observer verification loop—not merely increased computation—drives the accuracy gain.

## G. Impact of Entropy-Skipping Threshold $\delta$

When the thinker's prior entropy $H_t$ falls below a threshold $\delta$, the observer call is skipped entirely, trading a small accuracy loss for a significant cost reduction. Table 8 presents the accuracy and cost trade-off on V* (beam width $K = 2$, depth $D = 3$) as $\delta$ is swept.

*Table 8.* Entropy-skipping ablation on V* (beam width $K = 2$, depth $D = 3$). "No skip" is the full V-ABS baseline; $\Delta$ is the accuracy difference relative to the no-skip baseline.

| Threshold $\delta$ | API Calls | Avg Tokens | Time (s/sample) | V* Acc | $\Delta$ |
|---|---|---|---|---|---|
| No skip | 49.1 | 55.8K | 2.78 | 89.01% | — |
| $\delta = 1.83$ | 47.9 | 54.2K | 2.74 | 89.01% | +0.00 |
| $\delta = 1.84$ | 45.9 | 53.3K | 2.62 | **90.05%** | **+1.04** |
| $\delta = 1.85$ | 45.6 | 53.0K | 2.60 | 89.53% | +0.53 |
| $\delta = 1.90$ | 40.7 | 50.9K | 2.39 | 87.43% | −1.53 |

The optimal threshold $\delta = 1.84$ simultaneously reduces API calls by 6.5% and *improves* accuracy by +1.04%, because skipping the observer when the prior is already highly confident avoids noisy posterior feedback that can corrupt a correct thinker decision. Below $\delta = 1.83$, no calls are skipped and the result is identical to the baseline. Above $\delta = 1.90$, too many

observer calls are skipped and accuracy degrades by 1.53%. We adopt $\delta = 1.84$ as the default entropy-skip threshold in our final configuration.

## H. Ablation on the Choice of Positive/Negative Token Set

Both the thinker and observer scores aggregate log-probabilities over a predefined positive token set (e.g., {Yes, No}). Table 9 compares four token-set choices across V*, VisuoThink, and Jigsaw.

*Table 9.* Token set ablation. "All combined" merges {Yes, No, True, False, Correct, Incorrect} and their common casing variants, and is our default setting.

| Token Set | V* Acc | VisuoThink Acc | Jigsaw Acc |
|---|---|---|---|
| {Yes, No} | 89.01% | 45.6% | 79.1% |
| {True, False} | 70.16% | 31.7% | 34.1% |
| {Correct, Incorrect} | 67.02% | 27.1% | 12.0% |
| All combined (default) | **89.53%** | **45.8%** | **79.2%** |

Using only {Yes, No} already yields strong results, confirming that the models reliably route their positive/negative intent through these tokens. Alternative token sets ({True, False}, {Correct, Incorrect}) suffer large drops, especially on Jigsaw, because these tokens appear less frequently in the instruction-following fine-tuning regime and their log-probabilities are less calibrated. Merging all six token groups into the "All combined" set provides the best performance across all three benchmarks with only marginal gains over {Yes, No}, so it is adopted as the default without any loss of simplicity.

## I. Effectiveness of the Heuristic Score

The scoring function in V-ABS combines thinker prior $\mathcal{F}_{pri}$, observer posterior $\mathcal{F}_{obs}$, and an optional task-specific heuristic $\mathcal{F}_{heur}$. To isolate the contribution of the heuristic, we ablate it on V* (beam width $K = 2$) and compare against a no-search direct-VQA baseline.

*Table 10.* Heuristic ablation on V* (beam width $K = 2$). The heuristic definition for visual search is $\mathcal{F}_{heur} = 0.7 \times (1/(1 + 0.1 \times d)) + 0.3 \times r_{\text{area}}$, where $d$ is the current depth and $r_{\text{area}}$ is the area ratio of the crop.

| Configuration | V* Acc |
|---|---|
| Direct VQA (no search) | 85.34% |
| V-ABS with heuristic | 87.43% |
| **V-ABS w/o heuristic** | **90.58%** |

Removing the heuristic *improves* accuracy by +3.15% over the heuristic-equipped variant, and yields a +5.24% gain over direct VQA. The depth-and-size heuristic actively biases the search toward shallower, larger regions—which is incorrect when the target is small and located deep in the hierarchy. Without the heuristic, the thinker-actor-observer loop alone achieves 90.58%, demonstrating that the accuracy gains reported in the main paper are attributable entirely to the general observer-verification principle rather than domain-specific engineering. Based on this finding, we recommend omitting task-specific heuristics by default and adding them only when strong domain prior knowledge explicitly justifies a bias toward certain state properties.

## J. Qualitative Visualizations

We provide qualitative examples to illustrate the inference process of V-ABS across diverse reasoning domains. In the visual search task shown in Figure 4, the model overcomes resolution bottlenecks by actively executing cropping operations. For instance, to identify the color of a parachute or a dustpan, the model iteratively crops specific quadrants (e.g., Top-Right "TR", Top-Left "TL"), progressively narrowing the search space until the target is resolved. Figure 5 illustrates the qualitative performance on the visual navigation and logic tasks. In the navigation, the agent performs sequential path planning on the frozen lake and VisuoThink benchmarks. At each step, the model predicts a move and physically updates the agent's

coordinates on the map. In the jigsaw puzzle task, the model restores scrambled images by predicting the correct permutation indices. The visual state is updated by physically rearranging the patches, allowing the observer to verify semantic coherence. In the Sudoku task, our technique solves the confusion through iterative filling. It generates the updated board state after inserting new digits (marked in blue), ensuring that each step satisfies the row, column, and box constraints.

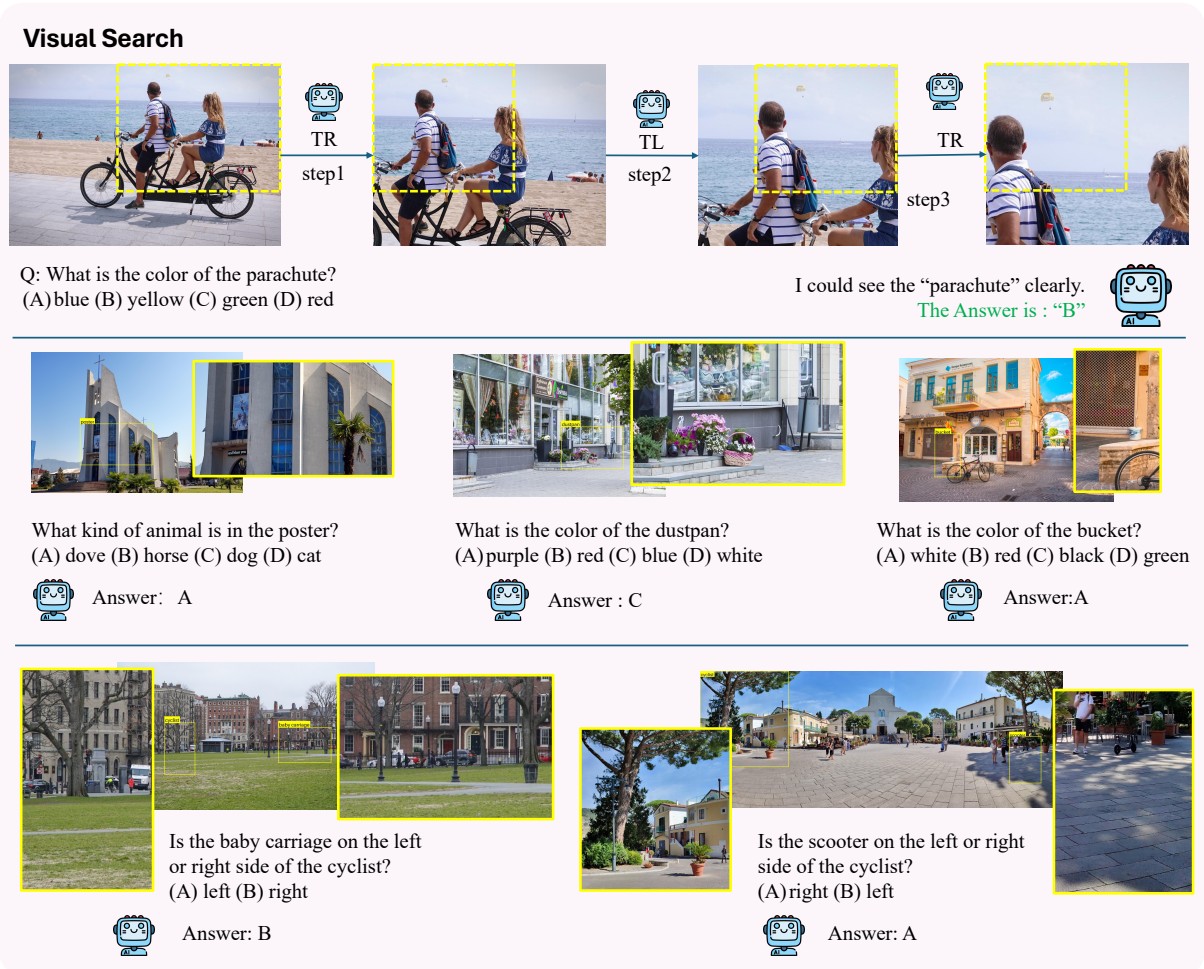

*Figure 4.* Qualitative results for Visual Search. The model actively crops specific regions (e.g., steps TR → TL) to locate small targets (parachute, dustpan) or verify spatial relations, addressing the resolution bottleneck.

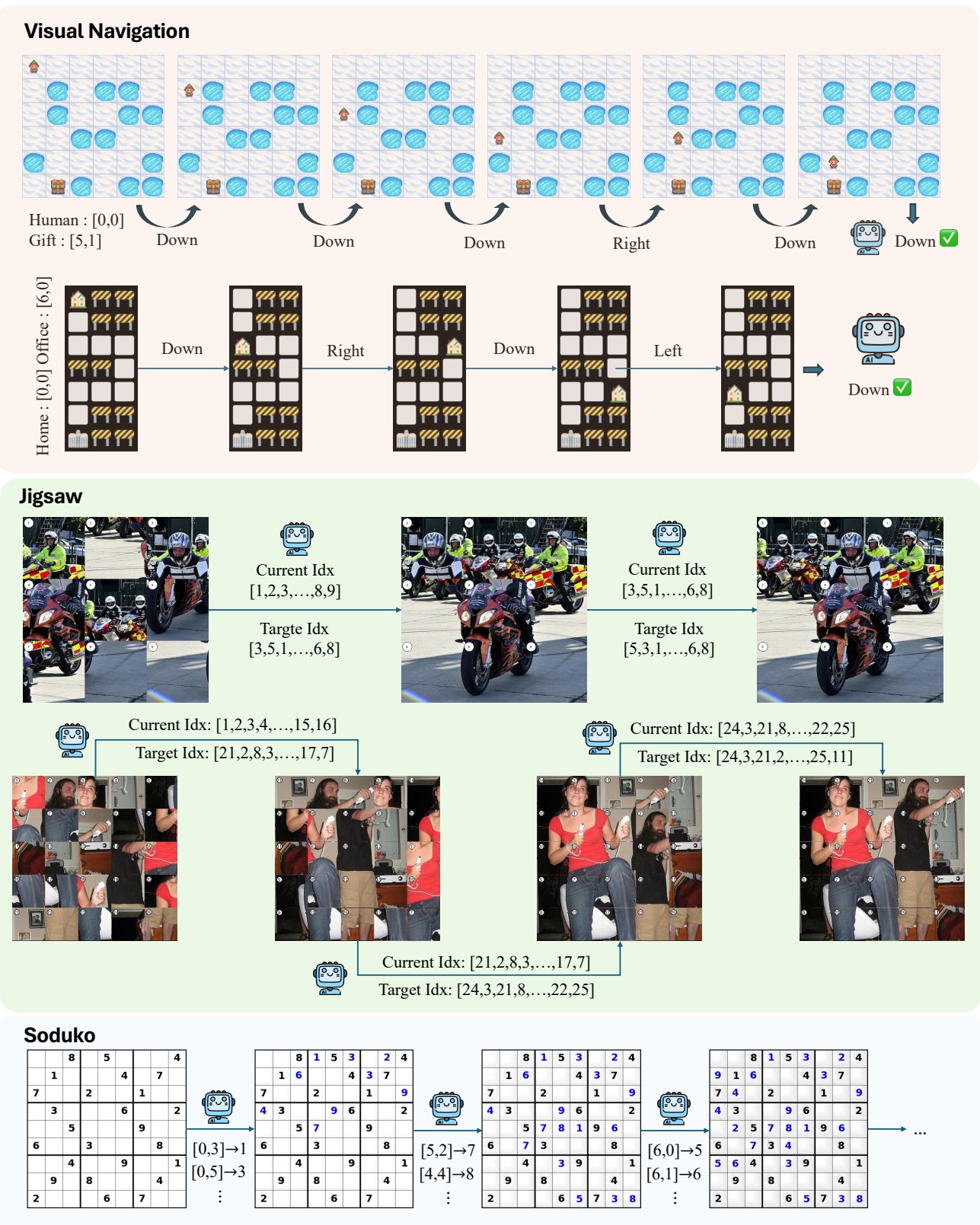

*Figure 5.* Qualitative results for Navigation, Jigsaw, and Sudoku. (Top) Visual Navigation: The agent iteratively updates its position on the map (Frozen Lake and VisuoThink), executing directional actions to reach the goal while avoiding obstacles. (Mid) Jigsaw Puzzle: The model restores spatial structure by predicting permutation sequences for scrambled patches (e.g., $3 \times 3$ motorbikes, $4 \times 4$ portraits) and physically rearranging the image. (Bottom) Sudoku: The model performs logical deduction by filling missing numbers step-by-step (visualized in blue), updating the board state to verify validity against game constraints.

