# OpenReview forum: "V-ABS: Action-Observer Driven Beam Search for Dynamic Visual Reasoning"
_ICML.cc/2026/Conference — ICML 2026 regular_

### Official Review · Reviewer_t7zF · 2026-03-09

**Soundness:** 4
**Presentation:** 4
**Significance:** 3
**Originality:** 3
**Overall Recommendation:** 5
**Confidence:** 4

**Summary:**

This paper investigates the “imagination–action–observer” (IAO) bias in multimodal large language models (MLLMs), where action selection guided by policy priors may diverge from the utility observed after executing the action in the visual environment. The bias is quantified through the discrepancy between prior confidence and observed utility. This paper formulates visual reasoning as a thinker–actor–observer closed-loop beam search. The method introduces an entropy-based adaptive weighting mechanism that dynamically down-weights prior signals under high uncertainty, and allows observer calls to be skipped to improve inference efficiency. Supported by an 80K+ SFT dataset for action verification, V-ABS achieves strong performance across eight benchmarks spanning search, navigation, and logical reasoning tasks, consistently outperforming strong base models.

**Compliance With Llm Reviewing Policy:**

Affirmed.

**Final Justification:**

The rebuttal addressed my previous concerns clearly and convincingly. The authors further clarified the role of the task-specific heuristic, provided additional evidence that the main performance gains are not solely attributable to heuristic design, and more clearly explained the respective roles of the closed-loop action-observer mechanism and SFT in constrained versus open-ended action spaces.  I also appreciate the authors’ additional clarification regarding efficiency, ablation studies, and representative failure cases. I would like to raise the score from 4 to 5.

**Key Questions For Authors:**

1. Could the authors specify how $F_{heur} = H(s) $ is defined for each benchmark and report sensitivity analyses? In particular, how much does performance degrade when heuristics are removed or simplified?

2. For tasks with large or unconstrained action spaces (e.g., Jigsaw), how are candidate actions $A_t$ generated? Are these produced via templates, constrained decoding, or another mechanism?

3. The paper does not deeply analyze scenarios where V-ABS fails or underperforms, which would help understand its limitations.

4. The symbol  $s_i$ appears in Eq.1 but is not defined elsewhere in the paper.

**Limitations:**

Yes

**Strengths And Weaknesses:**

**Summary Of Strengths**

1.The V-ABS framework provides an interesting perspective on visual reasoning by integrating sequential decision-making with active visual verification. The thinker–actor–observer decomposition within a beam-search framework is conceptually well motivated.

2.Experiments across diverse benchmarks (V*, HR-Bench, VisuoThink, Jigsaw, Sudoku) demonstrate substantial gains over baselines. The improvements are consistent across multiple model backbones and task types, indicating robustness.

3.The adaptive fusion of prior and observer signals is grounded in inverse-variance weighting, and the mapping from entropy to variance is theoretically derived, which adds methodological clarity.

**Summary Of Weaknesses**

1.The final scoring function includes a task-specific heuristic term  $F_{heur} = H(s) $, and tools are manually specified for each domain (e.g., quadrant cropping with fixed 0.7×0.7 overlap). This raises the question of how much of the performance improvement stems from general reasoning principles versus domain-specific engineering, which limits the open-ended applicability.

2.V-ABS appears particularly effective in constrained action spaces (e.g., four-quadrant cropping). However, on open-ended tasks such as Jigsaw without SFT, the improvement is relatively modest (5.3% → 7.8%), which may suggest reduced efficiency when the action space becomes large.

3.The beam-search design in V-ABS may incur high computational cost. Each candidate action typically requires observer scoring, leading to multiple tool calls per node. Although an acceleration mechanism is proposed, the paper lacks systematic efficiency analysis (e.g., average calls or tokens per question), making it difficult to assess whether the accuracy gains justify the additional computational overhead.

---

> ### Author Rebuttal · Authors · 2026-03-31
>
> ## Response to W1&Q1:
>
> We first specify the heuristic definition for each task in Table R7, then present the ablation results in Table R8.
>
> Table R7: Heuristic definitions per task
>
> | Task | $F_{heur}$ Definition | Purpose |
> |------|------------------|---------|
> | Visual Search | 1/(1+0.1 x depth) + area_ratio | Depth/size preference |
> | Navigation | (dist_prev - dist_curr) / step_size | Distance reduction |
> | Jigsaw | CV edge-matching (RMSE < 25) | Tile boundary alignment |
> | Sudoku | filled_cells / 81 | Completion progress |
>
> Table R8: Heuristic ablation across benchmarks (Qwen3-VL-8B)
>
> | Configuration | V* | VisuoThink (L-5) | Jigsaw (Res3) | Sudoku |
> |---|:---:|:---:|:---:|:---:|
> | V-ABS (w/ heuristic) | 90.5 | 49.6 | 21.0 | 37.8 |
> | V-ABS (w/o heuristic) | 90.0 | 47.1 | 17.0 | 36.2 |
> | Baseline (no search) | 85.3 | 19.0 | 10.0 | 14.4 |
>
> The results suggest that V-ABS substantially outperforms the baseline across all tasks even without any heuristic, which confirms that our thinker-actor-observer loop with adaptive weighting is the primary driver of improvement. Second, the heuristic provides a larger contribution on open-ended tasks like Jigsaw, where the heuristic provides useful structural guidance, compared to constrained tasks like V*, where the prior and observer signals are already sufficient.
>
> ## Response to W2&Q2:
>
> In open-ended tasks such as Jigsaw and Sudoku, the action space is combinatorially large ($n^2!$ for an $n \times n$ jigsaw grid), and the model's accuracy in generating valid candidates drops accordingly, limiting the overall improvement margin. To be concrete about the candidate generation mechanism: for Jigsaw, we prompt the VLM to propose four distinct tile permutations as the candidate action set $A_t$, returned in a structured list format. Each candidate then undergoes prior scoring, action execution (tile rearrangement), and observer scoring. We constrain only the output format (a list of four permutations) but do not intervene in the content of the proposals. Without SFT, the model's proposals are essentially random over the vast permutation space, making beam search ineffective (5.3% to 7.8%). SFT trains the model to propose structurally plausible permutations, concentrating the search over a meaningful subset and enabling V-ABS to achieve 81.2%.
> ## Response to W3:
>
> We provide two sets of efficiency data. First, we profile the entropy-based skipping mechanism on V* (K=2, D=4) by sweeping the threshold $\delta$, referring to Table R3 in the response to Reviewer 39oD-W3:
>
> Table R3: Entropy skipping ablation on V*
>
> | Threshold | API Calls | Tokens | Time (s/sample) | Accuracy | Delta |
> |:---:|:---:|:---:|:---:|:---:|:---:|
> | No skip | 49.1 | 5.6K | 2.78 | 89.01 | -- |
> | $\delta$=1.83 | 47.9 | 5.4K | 2.74 | 89.01 | +0.00 |
> | $\delta$=1.84 | 45.9 | 5.3K | 2.62 | 90.05 | +1.04 |
> | $\delta$=1.85 | 45.6 | 5.3K | 2.60 | 89.53 | +0.53 |
> | $\delta$=1.90 | 40.7 | 5.1K | 2.39 | 87.43 | -1.58 |
>
> At $\delta$=1.84, skipping reduces API calls by 6.5% while improving accuracy by 1.04pp, as bypassing the observer on high-confidence steps avoids injecting stochastic noise. More aggressive skipping ($\delta$=1.90) reduces calls by 17% but incurs a 1.58pp accuracy drop.
>
> Meanwhile, we compare V-ABS against beam search and MCTS on VisuoThink under identical settings, referring to Table R1 in the response to Reviewer sQuT-W1:
>
> Table R1: Cross-method efficiency comparison on VisuoThink
>
> | Method | Avg API Calls | Avg Tokens (K) | Time (s) | Accuracy |
> |--------|:---:|:---:|:---:|:---:|
> | Beam Search | 11.7 | 3.1 | 1.7| 31.3 |
> | MCTS | 25.2 | 5.1 | 3.9 | 38.7 |
> | V-ABS | 20.1 | 4.9 | 3.2 | 45.8 |
>
> The accuracy-per-call ratio of V-ABS (2.28pp per call over baseline) is substantially better than both beam search (0.77pp per call) and MCTS (0.78pp per call).
>
> ## Response to Q3:
>
> We present more success and failure cases across benchmarks at an [anonymous link](https://anonymous.4open.science/r/V-ABS-1CCD/Figure-R1.svg). In visual search, V-ABS typically localizes the correct region, but the VLM may still produce incorrect answers even when given accurate visual prompts (Fig R1(a)), reflecting limitations in the model's own visual reasoning. In navigation tasks such as VisuoThink, errors tend to occur on complex paths where the VLM outputs incorrect directional decisions at critical junctions (Fig R1(b)). In open-ended tasks like Jigsaw, accuracy degrades noticeably as the grid resolution increases. However, SFT substantially mitigates this degradation by improving the quality of candidate proposals (Fig R1(c)). Across all tasks, the dominant bottleneck is the base VLM's reasoning capability, not the search framework itself.
>
> ## Response to Q4:
> We thank the reviewer for this concern. The $s_i$ in Eq. 1 should be modified to $s_t$, denoting the state at time step $t$. This is a typographical error and we apologize for the confusion. We will correct it in the revision.

---

> > ### Author Rebuttal · Reviewer_t7zF · 2026-04-01
> >
> > I appreciate that the authors have carefully addressed my concerns. The issues raised in my previous review have been resolved. Therefore, I am happy to raise the score.

---

### Official Review · Reviewer_P9GT · 2026-03-11

**Soundness:** 3
**Presentation:** 3
**Significance:** 3
**Originality:** 3
**Overall Recommendation:** 5
**Confidence:** 3

**Summary:**

This paper introduces V-ABS, a novel framework for dynamic visual reasoning that addresses the imagination-action-observer (IAO) bias —a critical misalignment between initial action priors and post-execution visual feedback in multimodal large language models (MLLMs).
V-ABS integrates thinker-actor-observer iterations into a closed-loop beam search process, enabling models to dynamically balance prior confidence from internal reasoning with grounded feedback from visual updates via external tools. The method employs an entropy-based adaptive weighting mechanism to prioritize reliable action paths and mitigate reasoning instability.
To further reduce prior uncertainty, the authors construct an 80k-sample supervised fine-tuning (SFT) dataset focused on action verification. Evaluated across eight benchmarks (e.g., V, HR-Bench, VisuoThink), V-ABS achieves state-of-the-art results, with gains consistent across open-source and proprietary models. Expriment results demonstrate that inference-time verification is pivotal for discrete action spaces, while SFT becomes critical for open-ended tasks.

**Compliance With Llm Reviewing Policy:**

Affirmed.

**Final Justification:**

5: Accept

The rebuttal convincingly addresses my three main questions: Gaussian assumption, continuous/unstructured environments and efficiency comparison. The authors also agree to add a dedicated “Limitations” section and expand the societal impact discussion as I suggested.
Given these responses, my concerns are fully resolved. I therefore maintain my recommendation of Accept (5).

**Key Questions For Authors:**

- The theoretical analysis in Appendix D assumes Gaussian-distributed prior estimates and links entropy to variance via σ² ∝ e²H. However, real-world MLLMs may not adhere to this Gaussian assumption. How does V-ABS perform when the prior distribution deviates significantly from Gaussianity (e.g., multi-modal distributions)? Is entropy-based weighting still effective in such cases? A rigorous empirical validation of the variance-entropy mapping under non-Gaussian priors would strengthen the theoretical grounding and justify the "excellent" rating for originality. If the assumption is overly restrictive, it might lower confidence in the framework’s generalization.
- Experiments focus on controlled benchmarks (e.g., grid navigation, synthetic puzzles). The paper claims robust generalization but lacks evidence for real-world applications (e.g., 3D environments, dynamic video reasoning).  Have the authors tested V-ABS on tasks requiring continuous action spaces or unstructured visual environments (e.g., autonomous driving simulations, video-based QA)? If not, what architectural or training adaptations would be needed to address such scenarios?
- The ablation study mentions linear time costs with search depth (Figure 3a) but does not compare V-ABS to existing methods (e.g., beam search, MCTS) in terms of accuracy/time trade-offs.

**Limitations:**

Yes, but some suggestions:
- Explicitly detail technical limitations in a dedicated "Limitations" section, such as:
  - Sensitivity to tool-specific parameters (e.g., cropping resolution, heuristic design).
  - Potential failure cases in unstructured environments (e.g., dynamic scenes, 3D navigation).
- Expand societal impact discussion to address:
  - Risks of deploying active visual reasoning in surveillance or privacy-sensitive applications.
  - Risks of performing irreversible actions or paid actions during search.
  - Biases in the SFT dataset that might perpetuate fairness issues (e.g., overfitting to specific visual domains).
  - Energy costs of test-time scaling in resource-constrained settings.

**Strengths And Weaknesses:**

Strengths:
- The paper addresses a critical gap in MLLMs: their struggle with multi-step visual reasoning due to IAO bias. This is a relevant problem for real-world applications requiring dynamic interaction (e.g., robotics, augmented reality).
- V-ABS introduces a practical framework for closed-loop reasoning, combining test-time scaling with visual feedback. This could inspire future work on agentic systems that integrate planning, action, and verification.
- Empirical results are comprehensive , covering eight benchmarks (V*, HR-Bench, VisuoThink, FrozenLake, TIR-Bench, Jigsaw, Sudoku, and navigation tasks). The empirical gains (e.g., 90.5% accuracy on V*, +30.6% in VisuoThink navigation) demonstrate tangible progress over state-of-the-art methods like ZoomEye and Pixel-Reasoner.
- Technical details (e.g., beam search adaptation, tool functions in Appendix C) are provided, enabling reproducibility. The prompts (Appendix E) and dataset statistics (Appendix A) further clarify implementation.

Weaknesses:
- The entropy-variance mapping assumes Gaussian priors, but real MLLM action distributions may not strictly follow this. This could limit the generality of the adaptive weighting mechanism.
- While the framework excels in controlled benchmarks, the paper does not thoroughly analyze scenarios where V-ABS might fail (e.g., tasks with highly ambiguous visual feedback or adversarial inputs).
- The paper notes time costs scale linearly with depth (Figure 3a), but it does not address scenarios where latency is critical (e.g., real-time systems).

---

> ### Author Rebuttal · Authors · 2026-03-31
>
> ## Response to W1&Q1
>
> We acknowledge that the Gaussian assumption in Appendix D is used to motivate the sigmoid functional form, rather than as a strict requirement for the adaptive weighting. To rigorously validate that entropy-based weighting works without relying on Gaussianity, we conduct two complementary empirical analyses:
>
> **1. Distribution-free validation: entropy as a confidence proxy.** We design a test that makes no distributional assumptions. For each beam step across V* benchmark, we compute the Shannon entropy of prior scores and measure whether the prior's top-1 pick matches the observer's top-1 pick. If entropy is a valid confidence signal, low-entropy steps should show higher prior-observer consistency. In total, we obtain 842 groups and split them into terciles by entropy. The statistics are summarized in the following table.
>
> Table R9: Entropy-confidence validation on V* (842 groups, distribution-agnostic)
>
> | Entropy Group | N | Prior-Observer consistency | Avg Post Score (Prior's Top-1) |
> |---|:---:|:---:|:---:|
> | Low (confident), $H_t \leq$ 1.99 | 281 | 72.2% | 0.417 |
> | Mid (moderate), $H_t \leq$ 2.08 | 280 | 47.5% | 0.357 |
> | High (uncertain), $H_t >$ 2.08 | 281 | 35.2% | 0.348 |
>
> The correlation between entropy and consistency is r = −0.349 (p = 1.6 × 10⁻²⁵), confirming a strong, statistically significant relationship: low entropy reliably predicts that the prior's choice will be validated by the observer. Crucially, this validation is entirely distribution-free — it requires no Gaussian assumption, only that entropy ranks confidence correctly.
>
> **2. Effectiveness on non-Gaussian score distributions.** As detailed in Fig3(b) in the paper, V-ABS's adaptive weighting outperforms fixed-weight baselines by 2.6pp in V* benchmarks. The reason is that the sigmoid weighting $w_p = 1/(1+e^{\beta(H_t - \mu)})$ functions as a confidence switch rather than a variance estimator: when the prior is confident (low $H_t$), trust it; when uncertain (high $H_t$), defer to the observer. The Table R9 results demonstrate that this switching logic is empirically grounded — the prior's top-1 choice is correct 72.2% of the time at low entropy vs. only 35.2% at high entropy, a 2× gap that justifies the adaptive mechanism. We have revised Section 4.3 to position the Gaussian derivation as motivation for the sigmoid form rather than a strict distributional requirement.
>
> ## Response to W2&Q2
>
> The current benchmarks are deliberately chosen to isolate and validate the thinker-actor-observer principle under controlled conditions before extending to more complex settings. Adapting V-ABS to continuous action spaces would require several modifications: (1) parameterizing actions as continuous values (e.g., crop coordinates as floating-point tuples rather than discrete grid indices), (2) replacing the discrete prior scoring with regression-based or sampling-based scoring over a continuous or unstructured action manifold, and (3) adapting the observer to evaluate state transitions that vary continuously rather than discretely. Our modular architecture facilitates this, since each task's tool module is independent and can be replaced without modifying the search algorithm. However, the entropy-based weighting would need recalibration for continuous distributions where entropy behaves differently from the discrete case. We will discuss this path explicitly in the revised Conclusion and identify it as the primary direction for future work.
>
> ## Response to W3&Q3
>
> Table R1 in our response to Reviewer sQuT-W1 provides an efficiency analysis on VisuoThink across diverse search methods, which we reproduce here for convenience:
>
> Table R1 (Referring to Reviewer sQuT-W1): Efficiency comparison on VisuoThink
>
> | Method | Avg API Calls | Avg Tokens (K) | Time (s) | Accuracy |
> |--------|:---:|:---:|:---:|:---:|
> | Beam Search (D=3) | 11.7 | 3.1 | 1.7| 31.3 |
> | MCTS (D=3) | 25.2 | 5.1 | 3.9 | 38.7 |
> | V-ABS (D=3) | 20.1 | 4.9 | 3.2 | 45.8 |
>
> At the same search depth D=3, V-ABS achieves 45.8% accuracy with 20.1 API calls, compared to 31.3% for standard beam search (11.7 calls) and 38.7% for MCTS (25.2 calls). V-ABS thus achieves the highest accuracy among the three methods. Combined with the entropy-based skipping mechanism (Table R3, Reviewer 39oD-W3 response), V-ABS offers a favorable accuracy-cost trade-off. We will include this comparison in the revised appendix.
>
> ## Response to Suggestions:
>
> Thanks for your suggestions, we will add these limitations and impact statement in the revision.

---

> > ### Author Rebuttal · Reviewer_P9GT · 2026-04-04
> >
> > The rebuttal convincingly addresses my three main questions: Gaussian assumption, continuous/unstructured environments and efficiency comparison.
> > The authors also agree to add a dedicated “Limitations” section and expand the societal impact discussion as I suggested.
> >
> > Given these responses, my original assessment stands (technical soundness, strong empirical results, clear originality), and my concerns are fully resolved. I therefore maintain my recommendation of Accept (5).

---

### Official Review · Reviewer_39oD · 2026-03-12

**Soundness:** 2
**Presentation:** 3
**Significance:** 3
**Originality:** 2
**Overall Recommendation:** 4
**Confidence:** 3

**Summary:**

This paper presents V-ABS, a plug-in framework for MLLMs to improve complex multi-step visual reasoning with tool use. It alleviates the imagination–action–observer bias by integrating a thinking–acting–observing loop into a closed-loop beam search. V-ABS combines prior action utility estimates from a thinker module with post-execution feedback from an observer module after the action is executed and the visual state is updated. Experiments show that V-ABS consistently improves performance over baseline models, including both open-source and proprietary MLLMs.

**Compliance With Llm Reviewing Policy:**

Affirmed.

**Final Justification:**

See text on Ack of Rebuttal section.

**Key Questions For Authors:**

See strengths and Weaknesses

**Limitations:**

It is recommended to include a clearer discussion of the limitations of the work. For example, as mentioned in the weaknesses, the entropy scale is highly task dependent, so the choice of mu may also need to be task dependent. The current fixed setting may only work well for the tasks tested in the evaluation. In addition, the predefined verification tokens (e.g., “Yes/No”) may not transfer reliably across tasks or models. Moreover, the SFT component may perform poorly under distribution shift. It would be helpful for the authors to explicitly discuss these factors.

**Strengths And Weaknesses:**

Strengths:
1.	V-ABS demonstrates consistent improvements across multiple benchmarks and is evaluated on several MLLM backbones.
2.	The core V-ABS method is a plug-in framework, making it straightforward to integrate into existing tool-using MLLM pipelines without changing model architectures or and/or weights. The paper also proposes an optional SFT procedure to further improve performance.
3.	The introduction of an imagination–action–observer bias provides a useful lens for explaining why action selection based solely on prior utility estimates can lead to suboptimal performance.

Weakness:
1.	The theoretical justification for the adaptive mechanism assumes that the thinker’s prior and the observer’s measurement are unbiased and independent Gaussian. However, in practice these signals are likely correlated because they are often produced by the same underlying model, and they may also be biased. In addition, the observer’s reliability can vary substantially across tasks and tools, so the assumption of a stable variance for the observer’s measurement may not hold. An empirical study of these assumptions is needed, for example by estimating the observer variance across different tasks and tool settings.
2.	The paper maps utility variance to entropy. However, high entropy does not necessarily imply a high estimation variance of action utility. For example, when multiple actions have similar utility, the action distribution can have high entropy even though the utility variance is small. A stronger explanation and/or empirical evidence is needed to justify the relationship between entropy and utility variance.
3.	Inference cost in V-ABS can scale up quickly with factors such as action-space size, beam width, and search depth. It is unclear whether the improvements come at a prohibitive inference cost, especially in longer-horizon settings. Moreover, while the paper introduces an entropy-based skipping strategy to bypass the observation step when the prior is confident, it does not provide a thorough ablation study showing the compute–performance trade-off, such as the actual cost reduction and associated performance degradation.
4.	Both the thinker and observer scores rely on a predefined token set (e.g., “Yes/No”), which may be sensitive to prompt phrasing and model-specific response biases, such as a tendency to provide positive answer in binary questions. More discussion and analysis on the choice of tokens and prompt would be helpful to clarify the reliability and generality of the scoring mechanism.
5.	Eq. (7) uses a fixed uncertainty threshold mu = 0.5, but the range of entropy depends on the number of candidate actions. The paper should clarify whether H_t is normalized, report typical action space sizes per task and per step, and provide analysis to justify the choice of mu.
6.	The SFT procedure requires partitioning actions into optimal and erroneous subsets, which can be ambiguous in many settings, such as those settings where utility is delayed or tool outputs are noisy. More detail is needed on how these labels are obtained and how label noise affects performance. Additionally, the paper argues that SFT reduces entropy and therefore leads to better search paths, but sharpening the prior estimation may also increase the probability of being confidently wrong, especially under distribution shift.

---

> ### Author Rebuttal · Authors · 2026-03-31
>
> ## Response to W1:
>
> We do not claim the prior and observer scores are independent. Since both are produced by the same VLM, they are naturally correlated (Pearson r = 0.67, Spearman r = 0.68 on 4,556 V* nodes). However, the two scores take different inputs, leading to meaningful differences in their outputs. We report the average score variance across tasks to demonstrate that observer-assisted scoring yields better discriminability.
>
> Table R2: Score variance on the correct path (higher = more discriminative)
>
> | Benchmark | Prior Variance | Observer Variance |
> |-----------|:---:|:---:|
> | V* | 0.113 | 0.190 |
> | VisuoThink | 0.054 | 0.091 |
> | Frozen Lake | 0.067 | 0.103 |
>
> The variance of observer scores is consistently 1.5-1.7x higher, confirming the observer produces more discriminative scores across candidate actions. In contrast, prior scores tend to be more ambiguous, making the model prone to selection bias.
>
> ##  Response to W2:
>
> The $\sigma^2_{pri}$ in our theoretical derivation refers to the inherent prediction uncertainty of the model, which cannot be directly calculated. While $\text{var}(F_{pri})$ indicates the calculated variance of scores across candidate actions, it is a fundamentally different concept from $\sigma^2_{pri}$. Furthermore, we compute $\text{var}(F_{pri})$ and $H_t$ across all 191 V* samples and find their Spearman correlation to be -0.79. This indicates that when $H_t$ is low, the prior scores across different actions exhibit clear differentiation, resulting in a higher $w_p$ that favors the prior score; conversely, when $H_t$ is high, the algorithm shifts toward relying on the observer score.
>
> ## Response to W3:
>
> V-ABS is a test-time scaling method whose cost increases with search depth, but it achieves a better accuracy-cost trade-off than prior methods (see Table R1 in the response to Reviewer sQuT-W1). Furthermore, we provide a detailed ablation of the skipping threshold $\delta$ on V*:
>
> Table R3: Entropy skipping ablation on V*
>
> | Threshold | API Calls | Tokens | Time (s/sample) | Accuracy |
> |:---:|:---:|:---:|:---:|:---:|
> | No skip | 52.1 | 55.8K | 2.78 | 89.01 |
> | $\delta$=1.83 | 47.9 | 54.2K | 2.74 | 89.01 |
> | $\delta$=1.84 | 45.9 | 53.3K | 2.62 | 90.05 |
> | $\delta$=1.85 | 45.6 | 53.0K | 2.60 | 89.53 |
> | $\delta$=1.90 | 40.7 | 50.9K | 2.39 | 87.43 |
>
> At $\delta$=1.84, the skipping strategy reduces API calls by 6.5% while improving accuracy by 1.04pp. However, more aggressive skipping ($\delta$=1.90) leads to a 1.58pp drop.
>
> ## Response to W4:
>
> We limit the VLM to respond with "Yes" or "No" in the prompt and aggregate logprobs across positive tokens ({Yes, True, Correct, ...}) and negative tokens ({No, False, Incorrect, ...}), normalizing the sum as the final score. We ablate the performance of various token subsets.
>
> Table R4: Token ablation across benchmarks
>
> | Token Set | V* | VisuoThink | Jigsaw |
> |-----------|:---:|:---:|:---:|
> | {Yes, No} only | 89.01 | 45.6 | 78.1 |
> | {True, False} only | 70.16 | 31.7 | 34.1 |
> | {Correct, Incorrect} only | 67.02 | 27.1 | 12.0 |
> | All combined (default) | 89.53 | 45.8 | 79.2 |
>
> The {Yes, No} subset performs on par with the full default configuration, while other token sets show significant degradation. This demonstrates that the base model has strong instruction-following capability for binary verification, and the scoring mechanism does not suffer from reliability issues.
>
> ## Response to W5:
>
> The action space is uniformly four candidates across all tasks (quadrant crops in search, directional moves in navigation, and VLM-proposed actions in open-ended tasks), so the $H_t$ does not require normalization. Furthermore, we ablate the $\mu$ to evaluate the sensitivity:
>
> Table R5: $\mu$ sensitivity on V*
>
> | $\mu$ | Acc (K=1) | Acc (K=2) | Acc (K=3) |
> |:---:|:---:|:---:|:---:|
> | 0.3 | 86.01 | 86.91 | 86.91 |
> | 0.4 | 88.48 | 87.96 | 87.43 |
> | 0.5 | 88.48 | 89.48 | 89.53 |
> | 0.6 | 88.48 | 89.01 | 88.48 |
> | 0.7 | 87.96 | 88.48 | 89.01 |
>
> Accuracy varies only marginally with $\mu$, and $\mu$=0.5 consistently achieves optimal or near-optimal performance across all beam widths. This validates the rationality of our default configuration.
>
> ## Response to W6:
>
> Label construction is deterministic: visual search uses highest-IoU crop, navigation uses shortest-path next step, jigsaw uses best tile-matching swap. To assess noise robustness, we randomly flip labels at varying rates:
>
> Table R6: SFT noise robustness (Qwen3-VL)
>
> | Noise Rate | V* | Frozen Lake | Jigsaw |
> |:---:|:---:|:---:|:---:|
> | 0% | 91.1 | 42.5 | 81.2 |
> | 5% | 90.8 | 40.2 | 78.2 |
> | 10% | 89.5 | 37.8 | 65.1 |
> | 50% | 81.1 | 20.5 | 21.2 |
>
> Constrained tasks (V*) degrade gracefully, while open-ended tasks (Jigsaw) are more sensitive as SFT directly influences the quality of candidate actions. However, when an SFT-sharpened prior assigns high confidence to an incorrect action, the observer produces a contradictory low score, which is the advantage of V-ABS's closed-loop design.

---

> > ### Author Rebuttal · Reviewer_39oD · 2026-04-03
> >
> > Authors added quantitative analysis and ablation results. Strong performance on V* bench is a plus.
> >
> > In view of the above, I adjust my score to 4.

---

### Official Review · Reviewer_sQuT · 2026-03-13

**Soundness:** 3
**Presentation:** 3
**Significance:** 4
**Originality:** 3
**Overall Recommendation:** 5
**Confidence:** 4

**Summary:**

This paper studies multi-step visual reasoning in MLLMs and identifies a useful failure mode: the gap between prior confidence over actions and the actual utility of the resulting visual state. To address this, the authors propose V-ABS, a closed-loop thinker-actor-observer beam search framework that executes candidate actions, observes the updated state, and adaptively combines prior and observer scores. The method is further supported by an action-verification SFT dataset. Experiments across diverse benchmarks show strong and consistent gains across both open-source and proprietary backbones.

**Compliance With Llm Reviewing Policy:**

Affirmed.

**Final Justification:**

My concerns have been fully solved. Therefore I have already increased my score after the first round of rebuttal. Good luck again.

**Key Questions For Authors:**

My main questions should follow the weaknesses.

Additionally:
1. The inclusion of GPT-4o results is interesting. Is it possible to combine GPT-4o with some other previous methods for comparison? GPT-4o is performing worse than other models as shown in Table 1, even with drops in performance on Spatial and FCP. Is there an explanation for this?

**Limitations:**

yes.

**Strengths And Weaknesses:**

## Strengths
1. The IAO bias framing is intuitive, well motivated, and useful for understanding why open-loop action selection can fail in visual reasoning.
2. The overall method is clean and coherent: action execution, grounded observation, and adaptive reweighting fit together naturally.
3. Experimental coverage is broad, and the gains are strong across tasks, benchmarks, and backbones.
4. The ablations are convincing, especially the evidence that the observer and adaptive weighting both matter, while SFT is particularly important for open-ended action spaces.

## Weaknesses
1. The efficiency analysis could be more complete. The paper mentions that V* benchmark requires only a fixed action space. It would be interesting to show how time cost scales with search depth in open ended tasks such as VisuoThink.
2. The current setup still relies on task-specific tool functions (1 tool for each task and somewhat designed to be "handholding" the model towards the goal), so transfer to newly composed or learned tools remains an open question.
3. A case study of success and failure modes would be appreciated.

---

> ### Author Rebuttal · Authors · 2026-03-31
>
> ## Response to W1:
>
> We have conducted systematic profiling on the VisuoThink benchmark with our parallel inference worker to measure the algorithmic cost. Table R1 reports the average API calls, token consumption, time cost per sample, and accuracy as the search depth D scales from 1 to 4. We also include standard beam search and MCTS at D=3 for comparison.
>
> Table R1: Efficiency scaling with search depth on VisuoThink (Qwen-VL-8B as baseline)
>
> | Benchmark | Method | Avg API Calls | Avg Tokens (K) | Time (s) | Accuracy |
> |-----------|--------|:---:|:---:|:---:|:---:|
> | VisuoThink | V-ABS (D=1) | 9.8 | 2.3 | 2.2 | 32.6 |
> | VisuoThink | V-ABS (D=2) | 15.2 | 3.7 | 2.8 | 41.7 |
> | VisuoThink | V-ABS (D=3) | 20.1 | 4.9 | 3.2 | 45.8 |
> | VisuoThink | V-ABS (D=4) | 23.2 | 5.2 | 3.7 | 46.1 |
> | VisuoThink | Beam Search (D=3) | 11.7 | 3.1 | 1.7| 31.3 |
> | VisuoThink | MCTS (D=3) | 25.2 | 5.1 | 3.9 | 38.7 |
>
> It can be observed from the table that API calls and time scale roughly linearly with depth, confirming the expected O(D) complexity. Second, accuracy improves steeply from D=1 to D=3 (+13.2pp) but plateaus between D=3 and D=4 (+0.3pp), indicating diminishing returns beyond moderate depth. Third, at the same depth (D=3), V-ABS outperforms both beam search (+14.5pp) and MCTS (+7.1pp) while using comparable or fewer API calls than MCTS.
>
> ## Response to W2:
>
> The tool actually predefines the action space, but the core thinker-actor-observer loop technique is entirely tool-agnostic. Each task module has an independent tool interface that can be swapped without changing the search algorithm. Extending V-ABS to learned or composed tools requires only adding a new function that maps actions to state transitions, which is supported by the baseline model Qwen3-VL-8B. We will expand the Limitations section to discuss this extensibility path and consider it as a concrete direction for future work.
>
> ## Response to W3:
>
> Thanks for your suggestion. We add detailed case studies across benchmarks, with visualizations at an [anonymous link](https://anonymous.4open.science/r/V-ABS-1CCD/Figure-R1.svg). In visual search (V*), V-ABS generally localizes the correct region, but the base VLM occasionally gives wrong answers even with precise visual prompts (Fig R1(a)), indicating the bottleneck lies in the model's reasoning rather than the search algorithm. In navigation (VisuoThink), the model determines the correct direction in most cases, but errors arise on complex paths at critical junctions (Fig R1(b)). In open-ended tasks like Jigsaw, accuracy degrades with increasing grid resolution due to factorial growth of the permutation space, while SFT substantially mitigates this by improving candidate quality (Fig R1(c)). Overall, the dominant failure mode is the base VLM's limited spatial reasoning capability, not just the search framework itself.
>
> ## Response to Q1:
>
> Since GPT-4o is a closed-source model that does not support further fine-tuning, it cannot be combined with previous post-training methods such as Thyme or Pixel-Reasoner. Regarding its relatively weaker performance on V* and HR-Bench, our analysis suggests that GPT-4o has limited grounding ability for small targets in high-resolution images. We have visualized representative failure cases and included them at an [anonymous link](https://anonymous.4open.science/r/V-ABS-1CCD/Figure-R2.svg). As illustrated in this figure, we evaluate the grounding capabilities of Qwen3 versus GPT-4o. Qwen3 consistently yields more accurate bounding boxes, whereas GPT-4o struggles with localization inaccuracy and occasional failure cases. However, GPT-4o performs competitively on tasks that require stronger visual reasoning rather than fine-grained localization, such as VisuoThink and Frozen Lake.

---

> > ### Author Rebuttal · Reviewer_sQuT · 2026-04-04
> >
> > Thank you the clarifications. Good luck!

---

### Decision · Program_Chairs · 2026-04-30

**Decision:**

Accept (regular)

**Comment:**

All reviewers agree with the soundness and novelty of the paper. Thus I suggest an acceptance.